# Autophagy acts as a brake on obesity-related fibrosis by controlling purine nucleoside signalling

Klara Piletic[1], Amir H. Kayvanjoo [2,3,8], Felix Clemens Richter [1,8], Mariana Borsa [1,8], Ana V. Lechuga-Vieco [1,8], Oliver Popp[2], Sacha Grenet [1,4], Jacky Ka Long Ko[5], Lin Luo[1,6], Kristina Zec [1], Maria Kyriazi[7], Harriet K. Haysom [1], Lada Koneva [1], Stephen Sansom [1], Philipp Mertins [2], Fiona Powrie [1], Ghada Alsaleh[1,7,9] & Anna Katharina Simon [1,2,9] ✉

A hallmark of obesity is a pathological expansion of white adipose tissue (WAT), accompanied by marked tissue dysfunction and fibrosis. Autophagy promotes adipocyte differentiation and lipid homeostasis, but its role in obese adipocytes and adipose tissue dysfunction remains incompletely understood. Using a mouse model, we demonstrate that autophagy is a key tissue-specific regulator of WAT remodelling in diet-induced obesity. Importantly, loss of adipocyte autophagy substantially exacerbates pericellular fibrosis in visceral WAT. Change in WAT architecture correlates with increased infiltration of macrophages with tissue-reparative, fibrotic features. We uncover that autophagy restrains purine nucleoside metabolism in obese adipocytes. This ultimately leads to a reduced release of the purine catabolites xanthine and hypoxanthine. Purines signal cell-extrinsically for fibrosis by driving macrophage polarisation towards a tissue reparative phenotype. Our findings in mice reveal a role for adipocyte autophagy in regulating tissue purine nucleoside metabolism, thereby limiting obesity-associated fibrosis and maintaining the functional integrity of visceral WAT. Purine signals may serve as a critical balance checkpoint and therapeutic target in fibrotic diseases.

Excess weight and obesity represent a major global health and socio-economic burden[1]. Obesity pathogenesis is characterized by a marked increase in white adipose tissue (WAT) mass, predominantly in subcutaneous and visceral locations, with the latter being more detrimental in obesity pathophysiology[2]. Excess adiposity is considered a major risk factor for metabolic complications, including type II diabetes mellitus and fatty liver disease[3]. While traditionally viewed as a highly specialized tissue for energy storage and mobilization, adipose tissue is now recognized as a dynamic endocrine and paracrine organ[4]. During excess nutrient availability, WAT mass increases through cell growth (hypertrophy) and number (hyperplasia) of adipocytes, which shift their metabolism to meet the energetic demands

[1]Kennedy Institute of Rheumatology, University of Oxford, Oxford, UK. [2]Max-Delbrück Center for Molecular Medicine in the Helmholtz Association, Berlin, Germany. [3]Immunology Program, Sloan Kettering Institute, Memorial Sloan Kettering Cancer Center, New York, NY, USA. [4]Master de Biologie, École Normale Supérieure de Lyon, Université de Lyon, Lyon, France. [5]Oxford-ZEISS Centre of Excellence, Kennedy Institute of Rheumatology, University of Oxford, Oxford, UK. [6]CAMS Oxford Institute, Nuffield Department of Medicine, University of Oxford, Oxford, UK. [7]Botnar Institute for Musculoskeletal Sciences, Nuffield Department of Orthopaedics, Rheumatology and Musculoskeletal Sciences, University of Oxford, Oxford, UK. [8]These authors contributed equally: Amir H. Kayvanjoo, Felix Clemens Richter, Mariana Borsa, Ana V. Lechuga-Vieco. [9]These authors jointly supervised this work: Ghada Alsaleh, Anna Katharina Simon. ✉e-mail: katja.simon@kennedy.ox.ac.uk

of the organism[5]. The obesity-associated metabolic shift predominantly includes core lipid and glucose metabolism to support energy storage and mobilisation, and these processes are tightly linked to functional mitochondria[6]. However, adipocyte metabolism and obesity-related metabolic rewiring beyond these pathways remain poorly understood. Adipose tissue architectural changes are supported by a dynamic remodelling of the extracellular matrix (ECM). Rapid and chronic expansion leads to hypoxia, chronic low-grade inflammation, and fibrosis, rendering WAT inflexible and dysfunctional[7]. Besides obesity-induced changes in adipocytes, adipose tissue dysfunction is also characterized by an accumulation of adipose tissue macrophages (ATMs)[8]. ATMs create an inflammatory milieu by releasing inflammatory cytokines and support adipose tissue fibrogenesis through ECM remodelling and fibroblast stimulation[9]. While the exact sequence of these processes is still unclear[10,11], their detrimental impact on adipose tissue is indisputable.

Autophagy is a fundamental process for the regulation of cellular metabolism and energy homeostasis[12]. Through a highly dynamic regulation of cellular recycling and degradation, autophagy controls metabolic adaptation, differentiation, homeostasis, and ultimately the overall function of cells and organs[13]. Autophagy is activated by various cellular and environmental stress signals, including nutrient and energy deprivation, and oxidative stress[14]. When initiated, it recycles organelles and macromolecules either as a quality control mechanism or to replenish energy and anabolic precursor pools. Through these processes, it can both rewire metabolic processes as well as supply nutrients, deeming it a master regulator of cellular metabolism[15,16]. Notably, autophagy can supply nutrients both in a cell-intrinsic as well as in a cell-extrinsic manner[17].

Autophagy supports adipocyte differentiation and lipid homeostasis, as well as facilitating communication between adipose tissue and the liver[18–21]; however, its function in obese adipocytes and adipose tissue dysfunction remains unclear and controversial[22,23]. Here, we demonstrate that in obese conditions, adipocytes upregulate autophagy to support their metabolic and structural adaptation. Failure to meet their metabolic demands in the absence of autophagy leads to elevated purine nucleoside production and release. Xanthine and hypoxanthine-mediated adipocyte-macrophage crosstalk drives a tissue-reparative macrophage phenotype and ultimately leads to excessive pericellular adipose tissue fibrosis.

## Results

### Autophagy is dysregulated in obesity and shapes WAT remodelling by limiting pericellular fibrosis

Adipocytes undergo significant structural, metabolic, and functional remodelling during obesity. To gain deeper insight into how obesity alters human WAT adipocytes, we re-analysed a recently published human WAT single-nuclei RNAseq (sn-RNAseq) atlas[24]. Comparison of white adipocytes between lean and obese states revealed macro-autophagy as one of the most notable, significantly dysregulated pathways, together with multiple pathways known to be impacted by weight gain, including insulin signalling, lipid metabolism, and tissue repair (Fig. 1A, B). Accordingly, we observed that in mice fed with high-fat diet (HFD) to induce obesity, autophagy initially increased alongside adiposity (Fig. 1C and Supplementary Fig. 1A). This observation correlated with previous reports of autophagy upregulation during obesity[22,25–30]. Prolonged HFD feeding, however, led to a significant downregulation of autophagic flux (Fig. 1C). To investigate its pathophysiological role, we generated a mouse model with an inducible, adipocyte-specific deletion of *Atg7* (*Atg7^Ad*) to circumvent defective adipogenesis. Specificity of Cre expression in adipose tissues and *Atg7* expression in metabolic tissues were assessed prior to this study[31]. *Atg7* encodes for an E1-like enzyme required for autophagosome formation, and its deletion leads to a profound loss of autophagic activity[13]. Through its ligase activity, *Atg7* is essential for the conjugation of LC3

to autophagic membranes, which is a critical step in the autophagy pathway[32]. Following the induction of *Atg7* deletion in mature adipocytes (Supplementary Fig. 1B), the obesity-associated increase in autophagy flux was abrogated in gonadal WAT (gWAT) adipose depot after the tamoxifen treatment, as assessed by western blot analysis at 16 weeks after HFD treatment (Supplementary Fig. 1C, D). Loss of autophagy in adipocytes had a profound impact on the tissue structure of obese *Atg7^Ad* gWAT (Fig. 1D). Obese *Atg7^Ad* mice showed exacerbated pericellular fibrosis compared to littermate controls (Fig. 1E, F). The onset of fibrosis correlated with an increase in obesity-induced gWAT autophagy in WT mice (Fig. 1C). Fibrosis in *Atg7^Ad* gWAT was obesity-dependent, as control (NCD)-fed *Atg7^Ad* mice did not develop fibrotic adipose tissue (Supplementary Fig. 1E, F). Pericellular fibrosis onset in gWAT developed during the initial body weight gain between six and nine weeks of HFD feeding (Fig. 1E, F). In line with the aggravated ECM accumulation, qPCR analysis revealed that several ECM components and enzymes, including *Col3a1*, *Fn1*, *Mmp14* and *Timp1*, were strongly increased (Fig. 1G). In WT mice, we noted that fibrosis significantly increased between 16 and 30 weeks of HFD feeding and then persisted over chronic HFD exposure up to 60 weeks (Supplementary Fig. 1G), suggesting that a decline in autophagy at 60 weeks may contribute to the maintenance of fibrosis. Taken together, these data point towards a critical protective role of obesity-induced autophagy in the control of ECM remodelling and tissue fibrosis.

### Multi-OMICS analysis reveals a key role for autophagy in adipocyte metabolic adaptation and nucleoside homeostasis during obesity

We next asked whether a striking shift in fibrotic processes in obese *Atg7^Ad* gWAT was due to an autophagy-mediated cell-intrinsic process. To explore the role of autophagy in adipocyte cellular remodelling, we conducted a multi-OMICS analysis of obese WT and *Atg7^Ad* adipocytes (Fig. 2). Proteomics analysis showed that classical autophagy receptors such as SQSTM1, which are typically degraded during autophagy, accumulated in autophagy-deficient adipocytes, confirming a lack of autophagy function (Supplementary Fig. 2A). Among the significantly enriched proteins (Fig. 2A), loss of adipocyte autophagy prominently altered metabolic processes, particularly nucleoside and lipid metabolism, as well as responses to oxidative stress (Fig. 2B, C and Supplementary Fig. 2A). Furthermore, the analysis highlighted impaired mitochondrial homeostasis in *Atg7^Ad* adipocytes, evidenced by reduced expression of the electron transport chain subunit complexes I-V following autophagy ablation (Supplementary Fig. 2B). Impaired metabolism in *Atg7^Ad* adipocytes was accompanied by a decrease in adipokine production and secretion, including leptin, adiponectin, and DPP4 (Supplementary Fig. 2C, D). To exclude that these findings could be solely explained by increased cell death of the autophagy-deficient adipocytes, we assessed adipocyte viability by Perilipin-1 staining, a widely used method in the field[33]. Clearly and as expected, adipocyte viability was markedly reduced due to HFD feeding, while autophagy depletion only increased the cell death minimally (Supplementary Fig. 2E, F). Perilipin-1 staining is based on morphological distinction of unilocular adipocytes and thus gives limited information about the type of cell death induced by loss of autophagy. To investigate whether it occurs via necrosis or programmed cell death (apoptosis), we also treated adipose tissue with the pan-caspase inhibitor Q-VD-OPh[34], which reduced adipocyte caspase 3 cleavage (Supplementary Fig. 2G, H), suggesting that at least some cell death occurred via caspase 3-induced apoptosis.

In line with our proteomics data, we found that adipocyte autophagy loss in obese mice profoundly impacted their cellular metabolism (Fig. 2D and Supplementary Fig. 3A). We observed that loss of autophagy led to a global reduction in both essential and non-essential amino acids, alongside enzymes supporting the TCA cycle and fatty acid oxidation in obese adipocytes assessed by

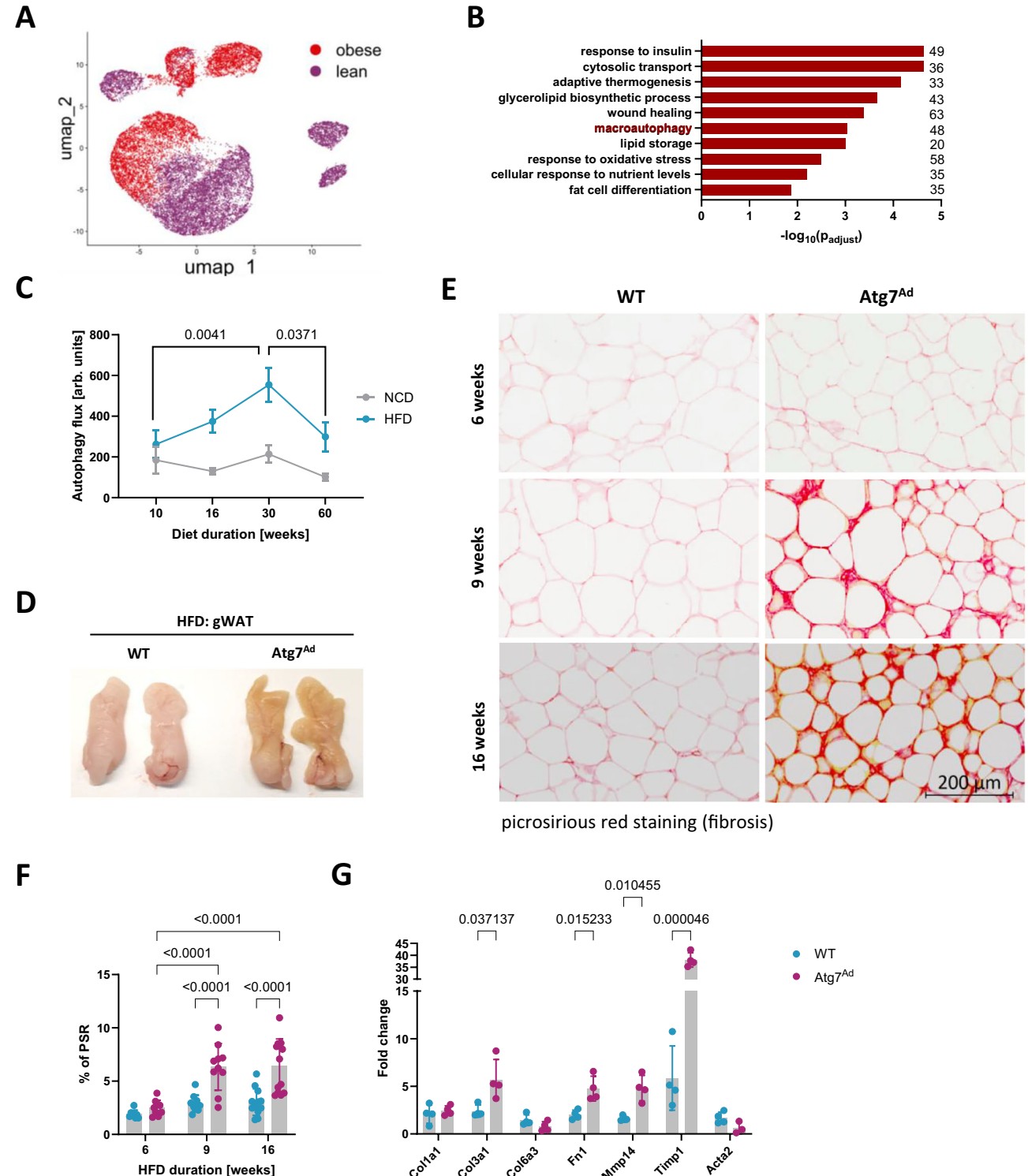

**Fig. 1 | Obesity dysregulates autophagy, which limits pericellular fibrosis in WAT. A** UMAP projection of human white adipocytes from lean (BMI < 30; 12822 adipocytes) and obese (BMI > 40; 9191 adipocytes) subjects. Single-nucleus RNA-seq data has been obtained from a deposited dataset (GSE176171). **B** Enrichment GO analysis of differentially regulated pathways in human adipocytes isolated from obese compared to lean WAT. Data analysis was conducted using the Seurat package v5.0.1. The number of genes identified for each term is labelled. **C** WT mice were fed a normal chow diet (NCD) or high-fat diet (HFD) for 10, 30 or 60 weeks before autophagy flux in gonadal white adipose tissue (gWAT) was assessed as explained in Materials and Methods. Western blot analysis of autophagy flux was calculated as (LC3-II (Inh) – LC3-II (Veh)). *n* = 3 (NCD-16), 5 (HFD-60), 6 (HFD-16), 7 (HFD10, HFD30), 8 (NCD-10, HFD-10) and 9 (NCD-30, NCD-60) mice. arb. = arbitrary

**D** Photograph of gWAT fat pads of WT and *Atg7^Ad* mice fed with high-fat diet (HFD) for 16 weeks. **E** Picrosirius red staining (PSR), specifically staining collagen I and III, of gWAT depots harvested from HFD-fed WT and *Atg7^Ad* mice after 6, 9 and 16 weeks of feeding. Representative images are shown. Scale bar, 200 μm. **F** Quantification of picrosirius red positive area as a percentage of total area from (**E**). *n* = 8 (WT-6), 10 (WT-9, *Atg7^Ad*-6, *Atg7^Ad*-9), 12 (WT-16) and13 (*Atg7^Ad*-16) mice. **G** Relative mRNA levels of ECM-related genes in gWAT after 16 weeks of HFD measured by qRT-PCR. *n* = 4 mice. Data are presented as mean ± SEM (**C**) or mean ± SD (**F**, **G**). Dots represent individual biological replicates. Data are representative (**D**, **G**) or merged from 3 independent experiments (**C**, **F**). Statistical analysis by two-way ANOVA with Tukey multi comparisons (**C**) or Fisher (**F**) test or multiple unpaired *t*-test (**G**).

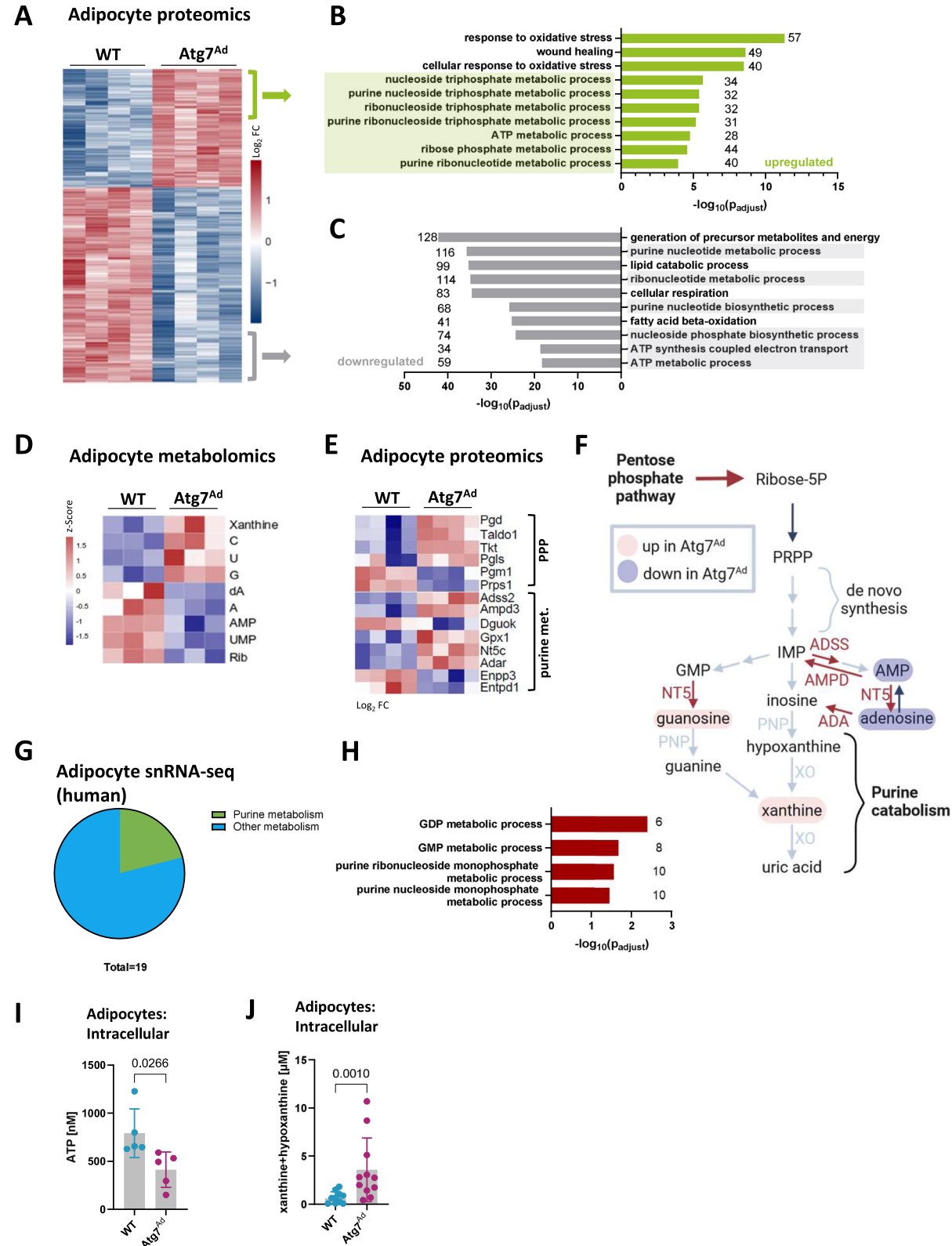

metabolomics and proteomics, respectively (Supplementary Fig. 3B–D). Furthermore, we found key components for RNA synthesis, including nucleotides UMP and AMP, adenosine, as well as ribose, significantly reduced (Fig. 2D).

Surprisingly, the loss of autophagy resulted in a pronounced upregulation of metabolites primarily associated with nucleoside metabolism (Fig. 2D). We found a strong accumulation of purine and

pyrimidine nucleosides in autophagy-deficient adipocytes, including guanosine, cytidine, uridine, as well as xanthine, a downstream product of purine catabolism (Fig. 2D). The dysregulation in nucleoside metabolism was further emphasised by altered protein levels of several critical enzymes involved in the pentose phosphate pathway (PPP) and intracellular purine metabolism (Fig. 2E). A prominent increase in PPP and purine metabolism enzymes was observed alongside elevated

**Fig. 2 | Autophagy controls adipocyte purine nucleoside metabolism.**
**A**–**C** Hierarchical clustering of proteomics profiles of enriched proteins in adipocytes isolated from gWAT of WT and $Atg7^{Ad}$ mice fed with HFD for 16 weeks (1886 proteins identified with adjusted $p$-value $\leq 0.01$). Colour-coding represents the $\log_2$ fold difference between WT and $Atg7^{Ad}$ mice (**A**). Enrichment GO analysis of differentially regulated pathways based on upregulated (**B**) or downregulated (**C**) proteins in adipocytes isolated from $Atg7^{Ad}$ compared to WT gWAT. The number of genes identified for each term is labelled. $n = 4$ mice. **D** Z-score heatmap of significantly ($p < 0.05$) abundant nucleotide and nucleoside metabolites in adipocytes. Metabolomics analysis was performed on adipocytes isolated from gWAT of WT and $Atg7^{Ad}$ mice following HFD feeding for 16 weeks. $n = 3$ mice. **E** $\log_2$ fold change heatmap of significantly differentially abundant proteins between WT and $Atg7^{Ad}$ involved in pentose phosphate pathway (PPP) and purine nucleoside metabolism in adipocytes as measured by proteomics analysis (as in **A**). **F** Schematic summary of adipocyte proteome and metabolome changes upon loss of autophagy depicting

simplified pentose phosphate and purine nucleoside metabolic pathways. Representative enzymes and metabolic products are colour-coded based on the fold change. **G** Pie chart representation of metabolic pathways identified in the enrichment GO analysis of differentially regulated pathways in human adipocyte snRNA-seq isolated from obese compared to lean WAT. **H** Differentially regulated GO biological pathways related to purine nucleoside metabolism in human adipocytes isolated from obese and lean subjects. The number of genes identified for each term is labelled. **I, J** Concentration of intracellular ATP (**I**) and xanthine and hypoxanthine (**J**) in gWAT adipocytes from WT and $Atg7^{Ad}$ mice fed with HFD for 16 weeks. $n = 5$ (**I**) and 11 (**J**) mice. Data are presented as mean ± SD. Dots represent individual biological replicates. Data are representative (**I**) or merged from 3 independent experiments (**J**). All heatmap values were scaled by row (protein/metabolite) using z-score. Statistical data analysis performed using the limma (v3.54.1) and clusterProfiler (v4.6.0) packages (**A**–**C**, **E**), one-way ANOVA (**D**), Seurat package v5.0.1 (**H**), unpaired $t$-test (**I**) or Mann–Whitney test (**J**).

glycolytic enzymes that support this metabolic axis (Supplementary Fig. 3D). Notably, these profound changes in adipocyte metabolism revealed that autophagy plays a critical role in the maintenance of functional nucleotide pools in adipocytes during obesity (Fig. 2F). Remarkably, when examining snRNA-seq data of human adipocytes, we found a similar pattern; out of 19 dysregulated metabolic pathways with obesity, more than one-fifth were related to purine metabolism (Fig. 2G, H). This parallel between mouse and human data underscores the central role of purine nucleoside metabolism in adipocytes and its regulation by autophagy during obesity.

To validate the role of autophagy in regulating purine metabolism, we measured both upstream and downstream intermediate metabolites with enzymatic assays, including ATP, hypoxanthine, xanthine, and uric acid. Obese $Atg7^{Ad}$ adipocytes showed a significant reduction in the energy-rich purine nucleotide ATP (Fig. 2I). In contrast, downstream intermediates of purine catabolism, xanthine and hypoxanthine, were significantly increased (Fig. 2J). However, there was no difference in the end product of purine catabolism, uric acid, or in the enzymatic activity of xanthine oxidase (Supplementary Fig. 3E, F), suggesting that purine catabolism was not upregulated to generate excess uric acid. Taken together, autophagy is indispensable to maintaining balanced purine nucleoside metabolism in obese adipocytes. Its impairment shifts the metabolism towards the increased catabolic activity of the nucleotide metabolic pathway, leading to the accumulation of downstream products, including nucleosides and purine bases.

**Autophagy limits obesity-induced xanthine and hypoxanthine release from adipocytes**

Based on the intracellular changes in nucleoside metabolism, we next investigated whether this has paracrine or endocrine effects on the tissue environment and systemically. Xanthine and hypoxanthine levels progressively accumulate in mouse serum over the time course of HFD feeding (Fig. 3A). Strikingly, serum xanthine and hypoxanthine were also found elevated in the absence of adipocyte autophagy (Fig. 3B), reflecting the intracellular metabolic rewiring in $Atg7^{Ad}$ adipocytes. Since we did not observe a shift of purine catabolism towards increased synthesis of its end product, uric acid, in $Atg7^{Ad}$ adipocytes, we postulated that xanthine might be, rather than converted to uric acid intracellularly, released into the extracellular milieu by adipocytes. To understand whether these nucleobases were adipocyte-derived and controlled by autophagy, we performed targeted metabolomics on the adipocyte secretome. Similar to their intracellular levels, we found a notable accumulation of cytidine, uridine, guanosine, and xanthine in the secretome derived from $Atg7^{Ad}$ compared to WT adipocytes (Fig. 3C). Xanthine and hypoxanthine levels more than doubled in the secretome of obese $Atg7^{Ad}$ adipocytes (Fig. 3D), and we observed a strong negative correlation between the activity of the autophagy pathway and their release from obese gWAT in WT and $Atg7^{Ad}$ mice (Fig. 3E). Purine nucleoside phosphorylase (PNP) plays a

key role in purine catabolism, limiting the production of purine nucleobases[35]. Inhibition of PNP activity by the clinically approved drug forodesine resulted in a significantly lower xanthine and hypoxanthine release from both WT and $Atg7^{Ad}$ adipocytes, with the latter being reduced to almost WT levels upon PNP inhibition (Fig. 3F). To exclude that increased apoptosis of $Atg7^{Ad}$ adipocytes is the reason for elevated nucleoside production and/or release, we inhibited gWAT apoptosis by pan-caspase inhibitor Q-VD-OPh or induced gWAT apoptosis by staurosporine (STS) (Supplementary Fig. 2G, H and Supplementary Fig. 3G, H). Neither treatment had an impact on xanthine and hypoxanthine efflux from WT or $Atg7^{Ad}$ adipocytes (Fig. 3G, H). However, when gWAT explants were lysed with 0.1% Triton-X, this resulted in markedly elevated extracellular levels in $Atg7^{Ad}$ explants (Fig. 3I). These results suggest that the mild increase in apoptosis due to loss of autophagy does not contribute to increased extracellular nucleoside levels. Furthermore, autophagy-depleted adipocytes actively generate xanthine and hypoxanthine through PNP activity that can be released when cells are fully lysed. Taken together, these assays suggest that while autophagy is not essential for efflux, it limits the excessive release of xanthine and hypoxanthine from visceral adipocytes during obesity.

**Adipocyte autophagy controls WAT remodelling by limiting immune cell expansion**

Given the dysregulated intra- and extracellular purine nucleoside signalling in $Atg7^{Ad}$ mice and the pronounced fibrosis, we set out to investigate whether these factors impact broader tissue remodelling and have a body-wide impact. We found that WAT body distribution exhibited remarkable differences with a reduced gWAT but expanded inguinal WAT (iWAT) deposition in obese $Atg7^{Ad}$ mice (Supplementary Fig. 4A). This was independent of weight gain or energy consumption between WT and $Atg7^{Ad}$ mice on either control (NCD) or HFD, where no differences were observed (Supplementary Fig. 4B, C). Deposition of fat into the visceral/gonadal area has been recognized to be more detrimental for obesity pathology, and it has been suggested that iWAT expansion could potentially buffer the deleterious effects of visceral fat increase[1,9]. In line with this hypothesis, changes in fat deposition in obese $Atg7^{Ad}$ mice were associated with improved obesity-induced metabolic syndrome, as demonstrated by increased glucose tolerance and lessened ectopic fat deposition in the liver (Supplementary Fig. 4D–G). Yet, no reduction in serum triglycerides and HDL cholesterol was observed (Supplementary Fig. 4H, I). In addition, obese $Atg7^{Ad}$ mice displayed no significant changes in circulating levels of adiponectin or leptin, as measured by ELISA, compared to controls (Supplementary Fig. 4J, K). Furthermore, autophagy-deficient adipocytes displayed no notable differences in cell size in gWAT (Supplementary Fig. 4L). These data suggest that autophagy-mediated adipocyte metabolic and tissue structural remodelling

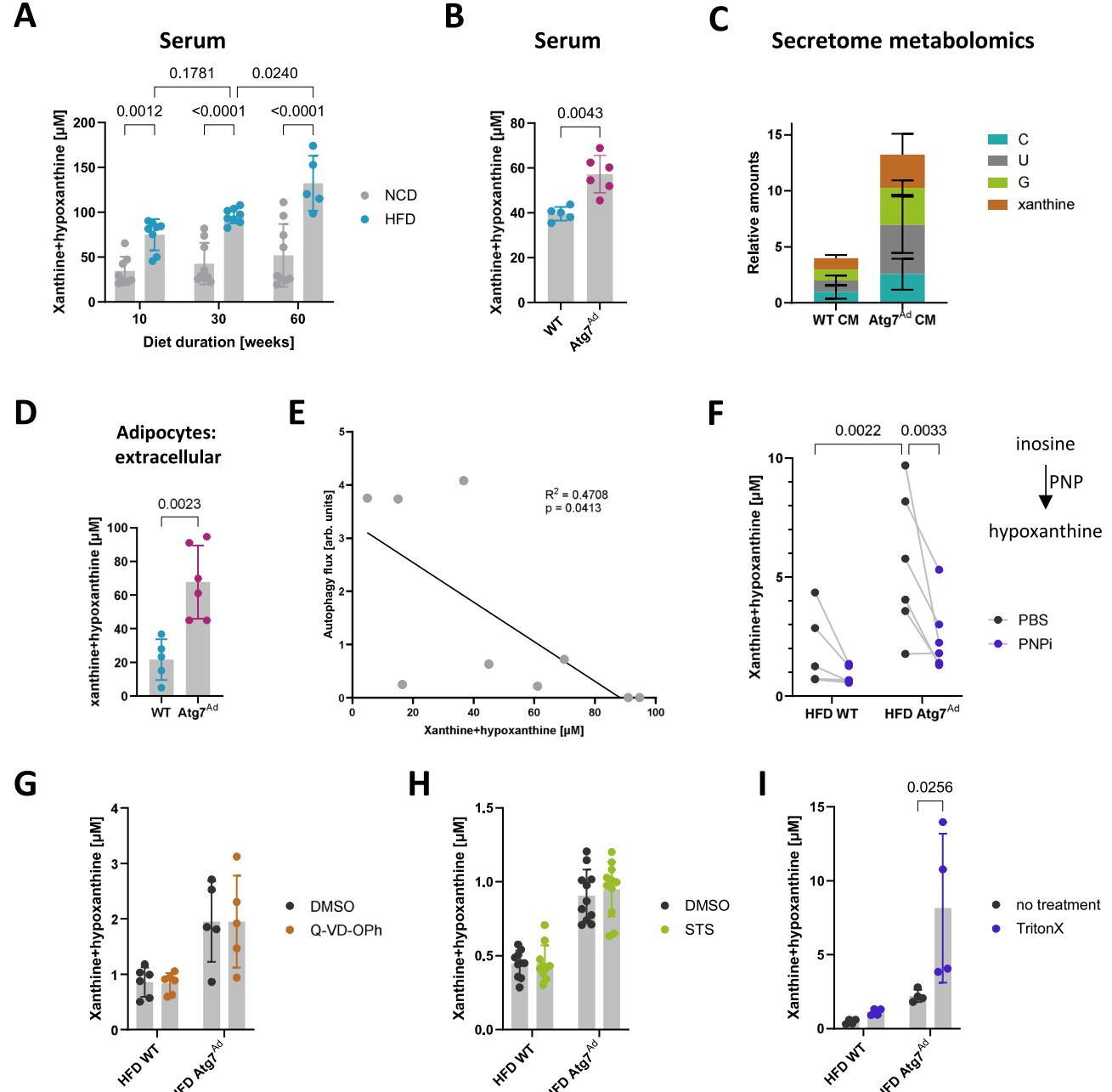

**Fig. 3 | Elevated release of xanthine and hypoxanthine in response to obesity by adipocytes is limited by autophagy. A**, **B** Concentration of serum xanthine and hypoxanthine in WT mice were fed NCD or HFD for 10, 30 or 60 weeks (**A**) or WT and *Atg7^Ad* mice fed with HFD for 16 weeks (**B**). *n* = 5 (HFD60, **B**-WT), 6 (**B**-*Atg7^Ad*), 8 (HFD10, HFD30, NCD10) and 9 (NCD30, NCD60) mice. **C** Relative abundance of nucleosides released by gWAT adipocytes isolated from WT and *Atg7^Ad* mice following HFD feeding for 16 weeks and measured in metabolomics analysis. *n* = 3 mice. **D** Concentration of xanthine and hypoxanthine released from gWAT adipocytes cultured over 24 h ex vivo. Adipocytes were isolated as in (**C**). *n* = 5 (WT) and 6 (*Atg7^Ad*) mice. **E** Correlation analysis of the level of autophagy flux in gWAT and concentration of xanthine and hypoxanthine released from gWAT adipocytes as in (**D**). Western blot analysis of autophagy flux was calculated as (LC3-II (Inh) – LC3-II (Veh)). *n* = 9 mice. **F** Concentration of xanthine and hypoxanthine as in (**D**) treated with either PBS or 10 μM forodesine, a purine nucleoside phosphorylase (PNP) inhibitor. *n* = 6 mice. **G** Concentration of xanthine and hypoxanthine released from gWAT explants cultured overnight ex vivo and treated with either DMSO or 20 μM Q-VD-OPh, a pan-caspase inhibitor. *n* = 5 (*Atg7^Ad*) and 6 (WT) mice. **H** Concentration of xanthine and hypoxanthine after treatment of gWAT explants with DMSO or 10 μM staurosporine (STS), an apoptosis inducer, for 24 h ex vivo. *n* = 11 mice. **I** Concentration of released xanthine and hypoxanthine from gWAT explants after cell lysis with 0.1 % Triton-X for 1 h before the end of incubation. *n* = 4 mice. Data are presented as mean ± SD. Dots represent individual biological replicates. Data are representative (**D**, **I**) or merged from 2 to 3 independent experiments (**A**, **B**, **E**–**H**). Statistical analysis by two-way ANOVA with Tukey multi comparisons (**A**), Mann–Whitney test (**B**), unpaired *t*-test (**D**), Pearson R correlation analysis (**E**), Fisher test (**F**) or Šídák multi comparisons test (**I**).

impact fat distribution in the pathological visceral WAT, alleviating obesity-induced metabolic syndrome.

Excessive ECM deposition in most tissues is commonly associated with increased secretion of pro-fibrotic cytokines that act to modulate the activity of ECM remodelling cells, such as transforming growth factor β (TGFβ) and osteopontin (OPN)[7,36,37]. To assess the production and release of these cytokines, we cultured gWAT ex vivo for six hours and measured their secretion by ELISA. Both TGFβ and OPN secretion were increased in *Atg7^Ad* obese mice (Fig. 4A, B). The increase in these two cytokines was associated with a significant elevation in nuclear

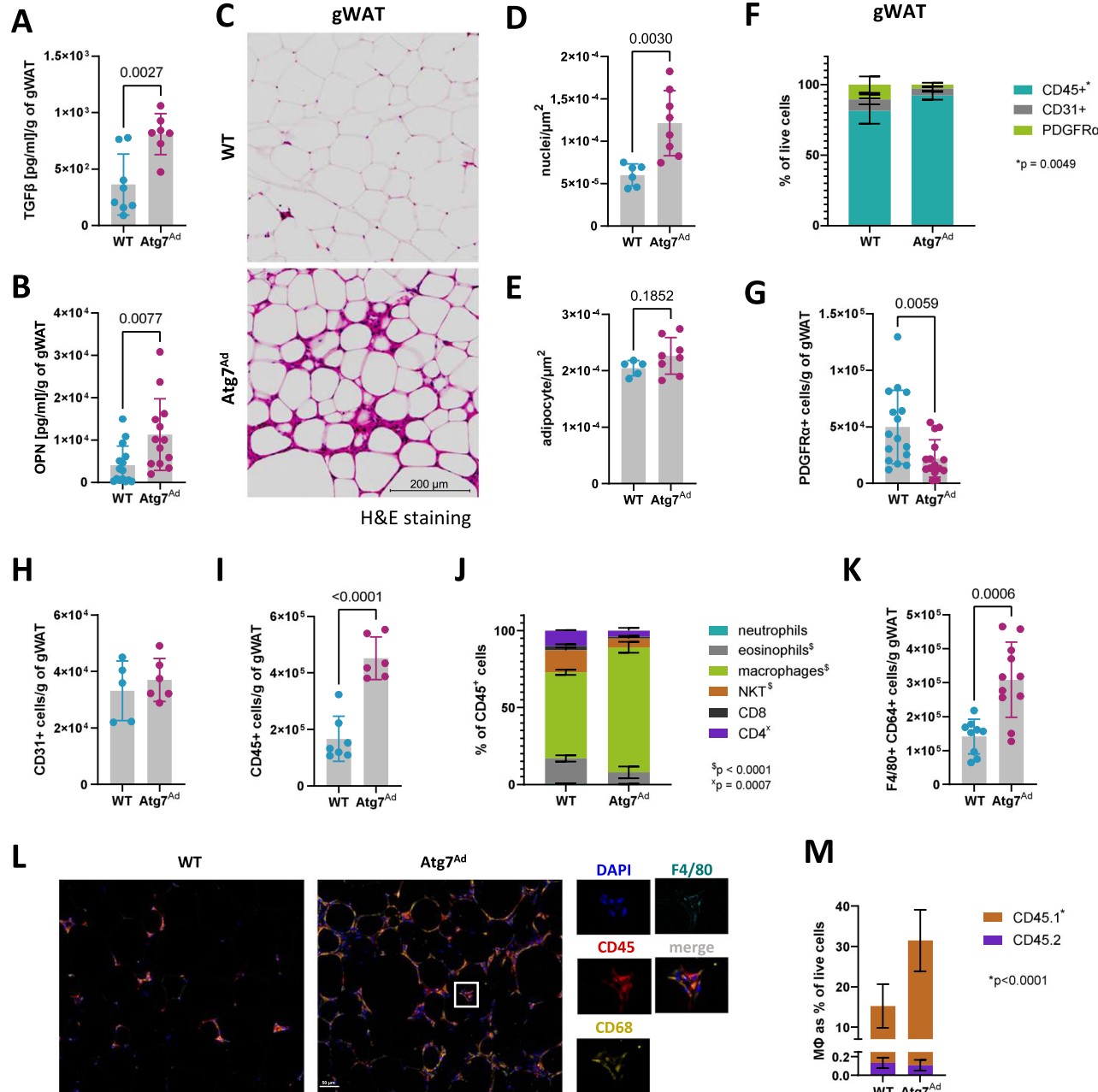

**Fig. 4 | Loss of adipocyte autophagy results in macrophage infiltration.** WT and *Atg7^Ad* mice were fed HFD for 16 weeks before gWAT was isolated for analysis. **A, B** Secretion of TGFβ (**A**) and osteopontin (OPN) (**B**) measured by ELISA. *n* = 7 (**A**-*Atg7^Ad*), 8 (**A**-WT), 13 (**B**-*Atg7^Ad*) and 15 (**B**-WT) mice. **C** H&E staining of gWAT. Scale bar, 200 μm. **D, E** Quantification of nuclei and adipocyte number from (**C**). *n* = 5 (**E**-WT), 6 (**D**-WT) and 8 (*Atg7^Ad* - **D**, **E**) mice. **F** Flow cytometry analysis of CD45⁺, CD31⁺, and PDGFRα⁺ populations in gWAT. *n* = 5 mice. **G–I** Absolute numbers of PDGFRα⁺ (**G**), CD31⁺ (**H**), and CD45⁺ (**I**) populations normalized to gram of WAT as in (**F**). *n* = 5 (**H**-WT), 6 (**H**-*Atg7^Ad*, **I**-*Atg7^Ad*), 7 (**I**-WT), 15 (**G**-*Atg7^Ad*) and 16 (**G**-WT) mice. **J** Flow cytometry analysis of immune cell (CD45⁺) composition. NKT natural killer T cell. *n* = 3 (WT) and 7 (*Atg7^Ad*) mice. **K** Flow cytometry analysis of F4/80⁺ CD64⁺ macrophage number in gWAT. *n* = 9 (WT) and 11 (*Atg7^Ad*) mice. **L** Representative immunofluorescence staining of F4/80, CD45 and CD68 of gWAT sections from WT and *Atg7^Ad* mice following HFD feeding for 16 weeks. **M** Flow cytometry analysis of F4/80⁺ CD64⁺ macrophage frequency labelled with CD45.1 (donor) and CD45.2 (host) congenic markers in gWAT after adoptive transfer. CD45.1 bone marrow cells were transferred in CD45.2 WT and *Atg7^Ad* hosts, where conditional knockout was induced 25 days following the transfer and mice were fed HFD for an additional 12 weeks. *n* = 6 mice. Data are presented as mean ± SD. Dots represent individual biological replicates. Data are representative (**C**, **F**, **J**, **L**) or merged from 2 to 3 independent experiments (**A**, **B**, **D**, **E**, **G–I**, **K**, **M**). Statistical analysis by unpaired *t*-test (**A**, **B**, **D**, **E**, **G–I**, **K**) or two-way ANOVA with Šídák multi comparisons test (**F**, **J**, **M**).

density in *Atg7^Ad* gWAT (Fig. 4C, D). This nearly threefold difference in nuclear density could not be attributed to adipocyte hyperplasia, as the number of unilocular adipocytes in the tissue remained constant (Fig. 4E). The fibroblast population (PDGFRα⁺), the main cell type involved in ECM dynamics, decreased in the *Atg7^Ad* gWAT (Fig. 4F, G). This observation was further supported by the expression of *Acta2*, a

gene largely restricted to myofibroblasts, which was not increased in gWAT of obese *Atg7^Ad* mice (Fig. 1G). Flow cytometry analysis also demonstrated that the endothelial cell population (CD31⁺) remained steady (Fig. 4H). Notably, the immune cell (CD45⁺) population, which can also be implicated in ECM dynamics[9], expanded significantly within the stromal vascular fraction of the *Atg7^Ad* gWAT (Fig. 4I).

Assessing the composition of the CD45[+] compartment in gWAT by flow cytometry revealed that macrophages were the most prevalent population upon obesity, which further increased in abundance in *Atg7[Ad]* mice (Fig. 4J, K and Supplementary Fig. 5). The increase in macrophage numbers was confirmed by in-situ imaging (Fig. 4L), with macrophage numbers more than doubling (Fig. 4K). To test whether these cells were derived from tissue-resident macrophages or infiltrating monocytes, we transplanted congenic bone marrow into WT or *Atg7[Ad]* hosts. Reconstitution of *Atg7[Ad]* mice with congenic CD45.1 bone marrow revealed that the majority of tissue-infiltrating macrophages were monocyte-derived, and of those, there were twice as many in the obese adipose tissue of the *Atg7[Ad]* host compared to the WT host (Fig. 4M). Collectively, these observations highlight the critical role of adipocyte autophagy in modulating the inflammatory environment during obesity by controlling macrophage infiltration.

### ATMs switch to a tissue-reparative phenotype in *Atg7[Ad]* gWAT

To better understand the identity and function of accumulated macrophages in *Atg7[Ad]* gWAT, we isolated F4/80[+] CD64[+] macrophages from gWAT of WT and *Atg7[Ad]* mice by fluorescence-activated cell sorting (FACS) and performed transcriptomics. Surprisingly, gene enrichment analysis of significantly dysregulated genes (Fig. 5A) revealed that loss of adipocyte autophagy induces downregulation of pathways associated with inflammation and cytokine production (Fig. 5B) while upregulating proliferative and tissue-remodelling processes as well as purine/nucleotide metabolism in ATMs (Fig. 5C). To validate the results, we first cultured ATMs from gWAT of WT and *Atg7[Ad]* mice ex vivo and measured their secreted cytokines. We confirmed that macrophages from obese *Atg7[Ad]* mice notably decreased their cytokine production of IL-1β, IL-6, TNFα, and IL-10 (Fig. 5D–G). In contrast, ATMs from obese *Atg7[Ad]* mice increased transcription of key pro-fibrotic tissue remodelling genes *Col3a1* and *Mmp14* (Fig. 5H).

The growing recognition of ATM plasticity and heterogeneity has revealed a complexity that renders the traditional M1/M2 (pro- and anti-inflammatory) paradigm overly simplistic and outdated[38,39]. The recent classification obtained under both normal chow and HFD using single-cell RNA sequencing (scRNA-Seq) suggests three main macrophage subtypes, including perivascular-like macrophages (PVM), non-perivascular-like macrophages (NPVM), and lipid-associated macrophages (LAM)[40–44]. While NPVMs and LAMs mediate inflammatory processes, PVMs control tissue repair[8]. Analysis of these macrophage populations by flow cytometry (Fig. 5I, J) revealed no difference in LAM (marked as F4/80[+] CD64[+] CD9[+] CD63[+]) abundance between obese *Atg7[Ad]* and WT gWAT (Fig. 5I, K). In contrast, we found tissue-reparative PVM (marked as F4/80[+] CD64[+] Lyve1[high] MHCII[low]) more than sevenfold increased and antigen-presenting NPVM (marked as F4/80[+] CD64[+] Lyve1[low] MHCII[high]) threefold decreased among macrophages isolated from *Atg7[Ad]* gWAT (Fig. 5J, L, M). In contrast to gWAT, iWAT displayed no increase in F4/80[+] CD64[+] ATM or Lyve1[high] MHCII[low] PVM accumulation, suggesting that macrophage abundance and phenotypes notably differ between the depots in *Atg7[Ad]* mice (Supplementary Fig. 6A, B). A significant, but much lower release of xanthine and hypoxanthine was observed from *Atg7[Ad]* iWAT adipocytes compared to gWAT (Fig. 3D and Supplementary Fig. 6C). This further underscored the intrinsic differences between these depots. In summary, we uncovered that in gWAT of obese *Atg7[Ad]* mice, macrophages switch from a predominantly pro-inflammatory to a tissue-reparative pro-fibrotic phenotype, which is accompanied by a strong ECM transcriptional signature.

### Metabolic dysregulation of *Atg7[Ad]* adipocytes is signalled through xanthine and hypoxanthine to macrophages for a tissue-reparative phenotypic switch

Observing that autophagy significantly impacted adipocyte purine nucleoside metabolism, which might, in turn, influence the surrounding microenvironment, we aimed to determine whether purine nucleosides could induce a tissue-reparative phenotype in macrophages. In pursuit of this goal, we first tested whether the adipocyte secretome could switch macrophages in vivo by cultivating ATMs isolated from lean adipose tissue in the presence of the secretome derived from either obese WT or *Atg7[Ad]* adipocytes. Three days after the exposure, we observed a significant increase in the Lyve1[high] MHCII[low] tissue repair macrophage population as well as the upregulation of ECM-related genes *Col3a1*, *Mmp14*, and *Timp1* in macrophages exposed to *Atg7[Ad]* adipocyte-derived secretome (Fig. 6A–C). These results mimicked our observations in vivo, suggesting that adipocyte-derived soluble signals are responsible for the macrophage phenotype.

We next aimed to determine whether purine nucleosides could be responsible for these observations. To this end, lean ATMs were cultured in vitro for 72 h in 50 ng/ml of M-CSF supplemented with 100 μM of either adenosine, guanosine, hypoxanthine or xanthine (Fig. 6D). Xanthine, and to a lesser extent hypoxanthine, led to a significant upregulation of ECM-related genes, whereas adenosine and guanosine did not. To further test whether hypoxanthine and xanthine could indeed trigger a tissue-reparative switch, lean ATMs were treated in vitro with the secretome of obese WT adipocytes for 72 h, supplemented with a mixture of 100 μM xanthine and hypoxanthine each. We observed a marked increase in the pro-fibrotic signature genes *Mmp14*, *Col3a1* and *Timp1* (Fig. 6E).

To shed light on signalling pathways induced by xanthine and hypoxanthine in macrophages, we performed transcriptomic profiling of ATMs treated with xanthine or hypoxanthine in vitro. This revealed significant upregulation of MAPK and ERK signalling pathways, alongside enrichment in GTPase activity and Ras signalling transcripts (Fig. 6F, G). These pathways are commonly triggered by G-protein coupled receptors (GPCRs), including purinergic receptors[45]. ERK activation is associated with a reparative macrophage phenotype and fibrosis[46–48]. Notably, P2Y10, a purinergic GPCR and one of the most upregulated genes following xanthine/hypoxanthine treatment in our data, has been linked to RhoA activation and downstream ERK signalling[49]. These findings suggest that extracellular purines may reprogram macrophages via purinergic GPCR-mediated MAPK-ERK signalling.

Collectively, these results suggest that increased release of xanthine and hypoxanthine from adipocytes can promote a tissue-reparative switch in macrophages during obesity. While the release is autophagy-independent, autophagy controls the purine nucleoside metabolism in adipocytes. Dysregulation of this metabolic pathway leads to excessive nucleoside release, which in turn shifts the balance from tissue inflammation toward fibrosis.

## Discussion

In this study, we have identified autophagy as a major brake on WAT fibrosis. Combining a genetic model and dietary intervention with proteomic, metabolomic, and functional analyses, we uncovered a critical role of autophagy in supporting adipocyte metabolic needs during excessive growth, limiting purine nucleoside catabolism. By studying the nucleoside metabolic changes upon loss of autophagy, we identified (hypo)xanthine-driven adipocyte-to-macrophage crosstalk. Finally, our work revealed a critical role of autophagy in limiting WAT ECM pathological remodelling through a (hypo)xanthine-induced macrophage tissue repair phenotype.

Understanding the changes in autophagy activity in adipose tissues during obesity in both humans and mice remained elusive, despite numerous reports[22,23,25–30,50]. In addition, the lack of clarity on the mechanism and function of autophagy in obese WAT highlighted the complex and poorly understood role of autophagy. While it has been reported that adipocyte autophagy supports adipose tissue-liver crosstalk, contradictory conclusions were drawn in the different

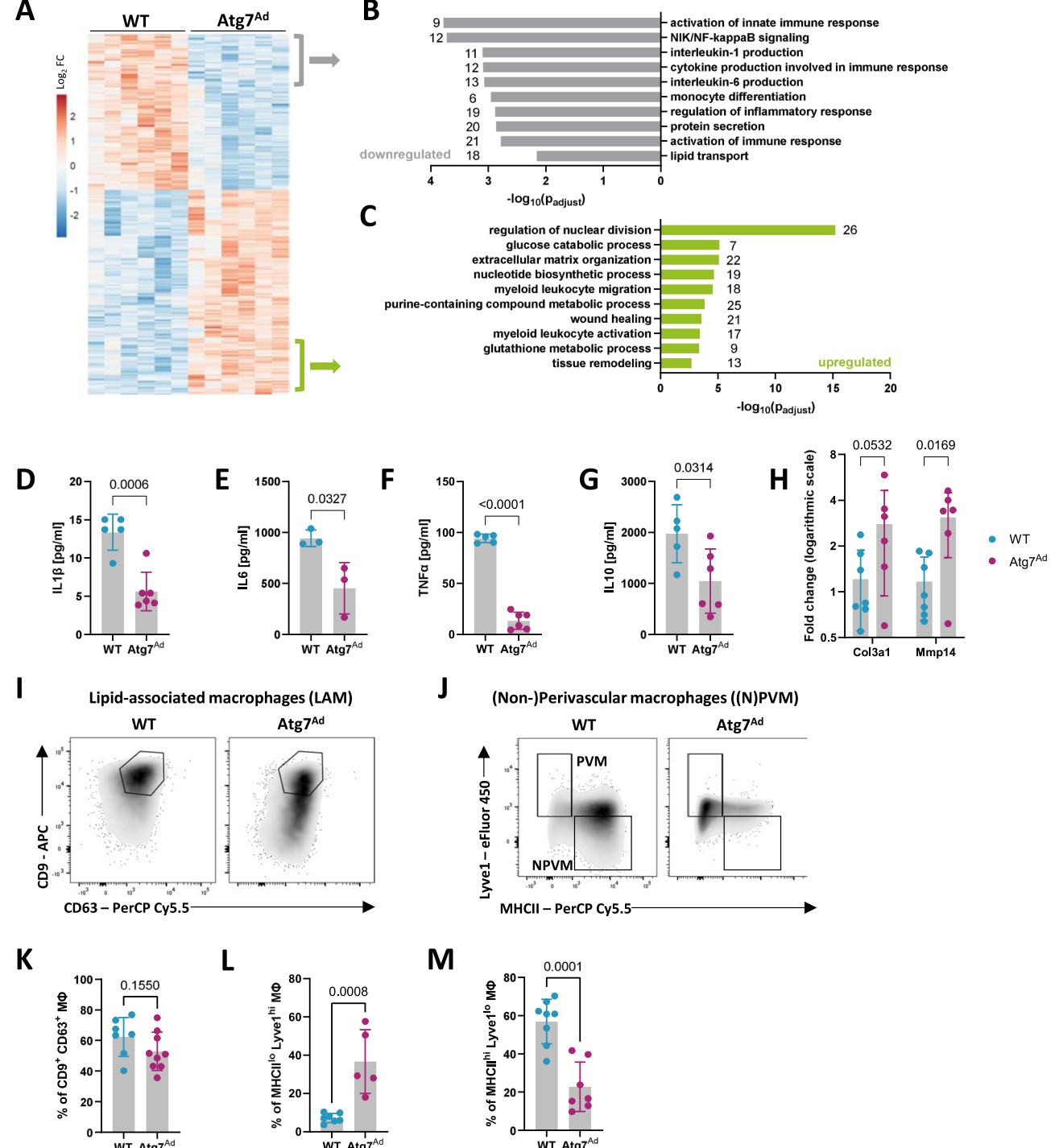

**Fig. 5 | Macrophages acquire a tissue-reparative phenotype upon autophagy loss in gWAT adipocytes.** **A**–**C** Transcriptomics analysis of F4/80⁺ CD64⁺ macrophages isolated from gWAT of WT and *Atg7Ad* mice fed with HFD for 16 weeks. Hierarchical clustering of transcriptional profiles of the top 1000 differentially expressed genes (**A**). Colour coding represents the log₂ fold difference between WT and *Atg7Ad* mice. Enrichment GO analysis of differentially regulated pathways based on downregulated (**B**) or upregulated (**C**) genes in macrophages isolated from *Atg7Ad* compared to WT gWAT. The number of genes identified for each term is labelled. *n* = 6 mice. **D**–**G** Secretion of IL-1β (**D**), IL-6 (**E**), TNFα (**F**), and IL-10 (**G**) by macrophages enriched from gWAT of WT and *Atg7Ad* mice fed HFD for 16 weeks. *n* = 3 (**E**), 5 (**D**, **F**, **G**-WT) and 6 (**D**, **F**, **G**-*Atg7Ad*) mice. **H** Relative mRNA levels of extracellular matrix (ECM)-related genes in sorted F4/80⁺ CD64⁺ macrophages

isolated from gWAT of WT and *Atg7Ad* mice fed HFD for 16 weeks measured by qRT-PCR. Data presented as log₂ fold difference. *n* = 6 (*Atg7Ad*) and 7 (WT) mice. Representative of 3 independent experiments. **I**–**M** Representative plots of lipid-associated macrophages (LAM) (**I**), identified as CD63⁺ CD9⁺, perivascular (PVM) and non-perivascular macrophages (NPVM) (**J**), identified as MHCIIlow Lyve1high and MHCIIhigh Lyve1low, respectively, assessed by flow cytometry. Quantification of LAM (**K**), PVM (**L**) and NPVM (**M**) frequency in the gates shown. *n* = 5 (**L**-*Atg7Ad*), 7 (**K**-WT, **L**-WT, **M**-*Atg7Ad*), 8 (**M**-WT) and 9 (**K**-*Atg7Ad*) mice. Data are presented as mean ± SD. Dots represent individual biological replicates. Data are representative (**D**–**G**) or merged from 2 to 3 independent experiments (**H**, **K**–**M**). Statistical analysis by the limma (v3.54.1) and clusterProfiler (v4.6.0) packages (**A**–**C**), unpaired *t*-test (**D**–**G**, **K**–**M**) or two-way ANOVA with Šídák multi comparisons test (**H**).

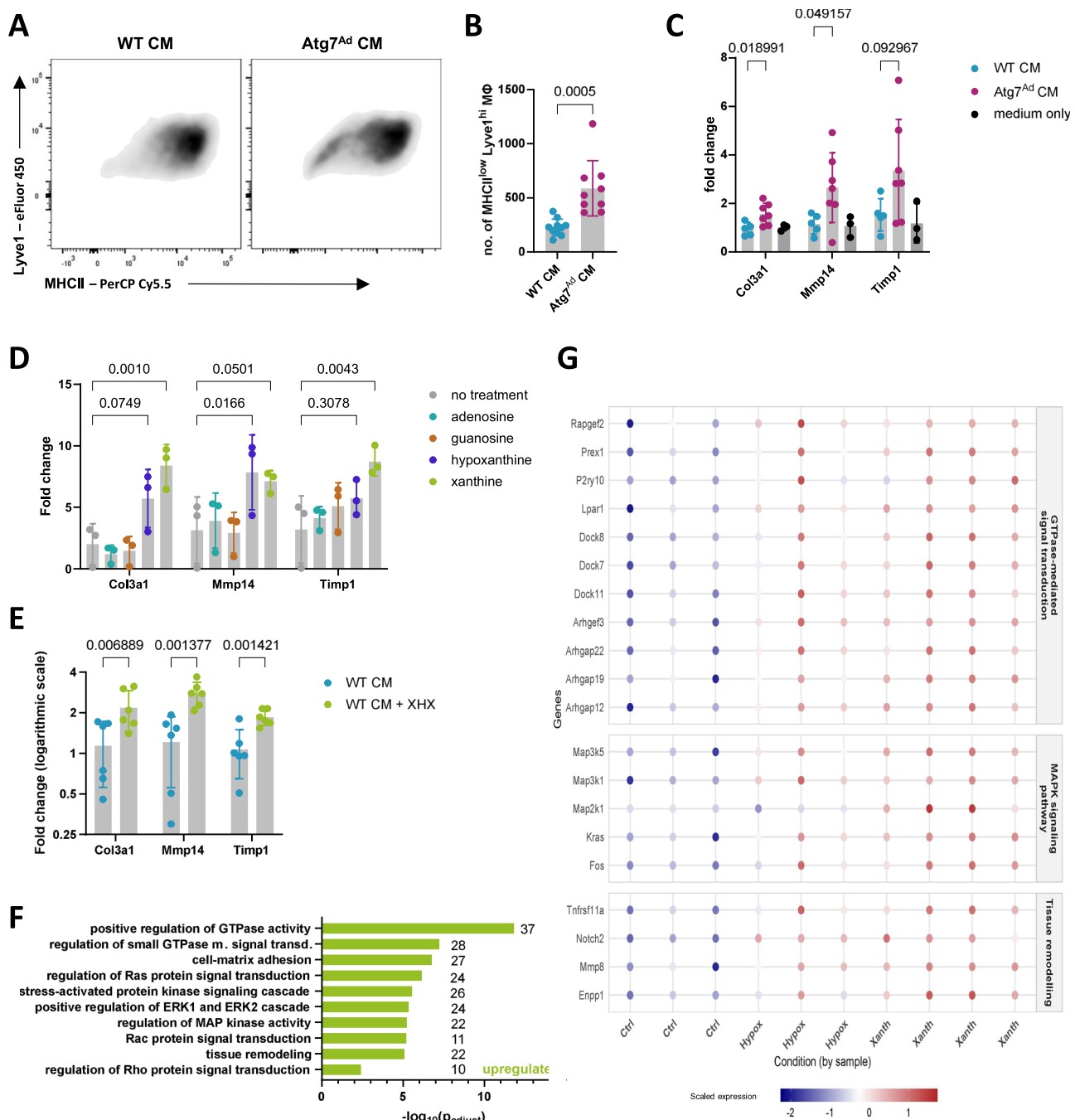

**Fig. 6 | Autophagy in obese adipocytes inhibits tissue-reparative macrophages and fibrosis via purine nucleoside signalling. A–C** Macrophages were isolated from lean WT gWAT and cultivated in vitro in the presence of conditioned medium (CM) generated by 24 h ex vivo incubation of obese gWAT WT and *Atg7^Ad* adipocytes. Representative plots of tissue repair macrophages assessed by flow cytometry (**A**). Quantification of flow cytometry analysis of MHCII^low Lyve1^high F4/80^+ CD64^+ macrophage number after 72 h of treatment with CM from WT or *Atg7^Ad* adipocytes (**B**). Relative mRNA levels of ECM-related genes in macrophages after 72 h of treatment with CM from WT or *Atg7^Ad* adipocytes or baseline full medium (**C**). RNA levels measured by qRT-PCR. *n* = 3 (**C**-medium only), 5 (**C**-WT), 7 (**C**-*Atg7^Ad*) and 9 (**B**) mice. **D** Macrophages were isolated from lean WT gWAT as in (**A–C**) and cultivated in vitro for 72 h in baseline full medium supplemented with 50 ng/ml of M-CSF and 100 μM of either adenosine, guanosine, hypoxanthine or xanthine. *n* = 3 mice. **E** Relative mRNA levels of ECM-related genes in macrophages as in (**A–C**)

with or without 100 μM supplementation of both xanthine and hypoxanthine (XHX). Data presented as log₂ fold difference. *n* = 6 mice. **F, G** Transcriptomics analysis of macrophages isolated from lean WT gWAT and cultivated in vitro in the presence of 50 ng/ml M-CSF and treated with 100 μM xanthine or hypoxanthine for 72 h. **F** Enrichment GO analysis of upregulated pathways in macrophages treated with either xanthine or hypoxanthine compared to the control based on shared DEGs. The number of genes identified for each term is labelled. **G** Dot plot of scaled expression of key DEGs regulating GTPase-mediated signal transduction, MAPK signalling pathway and tissue remodelling. *n* = 3 (Ctrl, Hypox) and 4 (Xanth) mice. Data are presented as mean ± SD. Dots represent individual biological replicates. Data are merged from 2 to 3 independent experiments (**B, C, E**). Statistical analysis by unpaired *t*-test (**B**), multiple unpaired *t*-tests (**C, E**), two-way ANOVA with Dunnett's multiple comparisons test (**D**), or the limma (v3.54.1) and clusterProfiler (v4.6.0) packages (**F, G**).

studies[20,21]. Our data suggest that obesity dysregulates autophagy both in humans and mice and that autophagy primarily increases with obesity in mice, with an eventual drop after prolonged HFD feeding, perhaps explaining a few studies that showed decreased autophagy levels with obesity[23,50]. We find that the primary function of autophagy is the support of the high metabolic demands of adipocytes during fat mass expansion. Adipocyte metabolism underlying fat storage and turnover is well understood[6], and is majorly determined by an increase in WAT mass. Nevertheless, our understanding of metabolic rewiring beyond glucose and lipid metabolism remains limited, with only scarce evidence for the role of other key metabolic processes in adipocytes[51–53]. We find that in humans, purine nucleoside metabolism represents one of the main dysregulated metabolic pathways in obese adipocytes. Our proteomics and metabolomics analyses revealed that autophagy critically supports nucleotide and amino acid pools in obese adipocytes. Similar autophagy-dependent changes have been previously observed in lung cancer cells under starvation and haematopoietic stem cells[54–56]. Similar to these cell types, mature adipocytes have a highly dynamic metabolic demand and can enter a pseudo-starvation state through adipokine signalling in obesity[57]. Furthermore, increased production of purine nucleoside catabolic intermediates, such as hypoxanthine and xanthine, has been previously suggested to relate to ATP depletion[54,58–60], which we also observed. Therefore, we believe these observations spanning several different cell types share a common molecular mechanism and highlight the indispensable role of autophagy in the provision of bioenergetic and biosynthetic substrates, responding to stress, maintaining redox homeostasis and survival.

Failure of adipocyte autophagy induction resulted in gWAT fibrosis, which is the more fibrosis-prone WAT depot[9]. The role of autophagy in fibrosis is controversial and highly context-dependent[61,62]. While the relationship between autophagy and adipose tissue fibrosis has not been experimentally addressed to date, their potential link has been proposed recently[63]. Tissue fibrosis develops when either ECM deposition or turnover become dysfunctional and is difficult to reverse[10]. Fibrosis of adipose tissue has been traditionally seen as detrimental as it mechanically stiffens the tissue, thereby negatively impacting its critical plasticity feature in the response to nutrient status[7]. Nevertheless, fibrosis is an essential component of tissue repair that limits tissue damage and aims to restore functional tissue architecture, improve recovery, and enhance survival[64]. In a chronic setting, when damage is persistent or severe, however, fibrosis leads to disruption of tissue architecture, interferes with organ function, and can ultimately lead to organ failure[65]. While chronic fibrosis occurs in gWAT of HFD-fed wild-type mice progressively, this fibrotic development was accelerated in adipose tissue deficient for adipocyte autophagy. The early fibrotic onset observed in *Atg7^{Ad}* gWAT, preceding elevation of autophagy flux in WT mice, suggests there is a demand for autophagy to maintain adipocyte cellular homeostasis early on during dietary challenge. This requirement for adipocyte autophagy is further exacerbated with prolonged HFD feeding, likely to reduce diet-induced cellular stress. We suggest that, initially, fibrosis acts to prevent acute and excessive tissue damage due to impaired adipocyte homeostasis and function upon autophagy depletion. Eventually, however, chronic accumulation of ECM likely leads to a broader adipose tissue dysfunction. Similar observations have recently been made in the pancreas[66]. Since WAT is not functionally compartmentalized, the detrimental effects of chronic fibrosis at the organismal level are difficult to discern. Indeed, increased deposition in the subcutaneous area positively correlates with more favourable disease outcomes compared to visceral deposition[3]. It has been proposed that the expansion of subcutaneous WAT could potentially help reduce the detrimental impact of visceral WAT expansion[9]. Concomitant with this, we observed the physical limitation of the pathological visceral WAT expansion by fibrosis,

improving glucose homeostasis, and reducing ectopic fat deposition in the liver. The role of autophagy in adiposity remodelling has also been observed in humans, where genome-wide association studies strongly linked *Atg7* to body fat distribution[67]. In addition, autophagy impairment, as seen in patients with biallelic MFN2 mutation (mitofusin 2, involved in mitochondrial fusion), was similarly found to impact depot-specific fat expansion[68,69]. Thus, a better understanding of autophagy as a determinant of WAT remodelling and fibrosis holds important therapeutic potential to improve obesity management and health outcomes of obese patients.

Excessive pericellular fibrosis positively correlated with a pronounced accumulation of tissue-reparative macrophages. Increased macrophage accumulation in adipocyte autophagy-deficient gWAT has been observed before, but never studied in detail[20,21]. In addition, caloric restriction studies in humans, known to induce autophagy, demonstrated reduced macrophage activation and, consequently, protection from adipose tissue fibrosis, but the link to autophagy was not made[70]. Macrophages are known as key regulators of tissue repair, regeneration, and fibrosis[71], and this may be true for adipose tissues as well[9,72]. The evidence, however, remains scarce, with elastin and TLR4 signalling being proposed to play a role in macrophage-induced WAT fibrosis during obesity[72,73]. On the other hand, ATMs have also been proposed to prevent pathological changes of ECM and limit the development of gWAT fibrosis[74]. Nevertheless, it remains unclear which signals induce the macrophage pro- or anti-fibrotic phenotypic switch that could serve as important balance checkpoints and therapy targets in fibrotic diseases. Local metabolic signals contributing to immune cell fates are becoming an area of increasing interest[75,76]. We show here for the first time that products of nucleoside catabolism, xanthine and hypoxanthine, can act as determinants of adipose tissue macrophage fate, resulting in a tissue-reparative phenotype. Our data suggest that this effect may be mediated via purinergic GPCR signalling, particularly through P2Y10, leading to activation of the MAPK-ERK pathway and downstream pro-fibrotic transcriptional programs. Notably, we found xanthine and hypoxanthine increased with obesity progression in mouse serum, and similar observations have been made before in humans, identifying adipose tissue as one of the main contributing factors[77–80]. Adipocytes have been previously described to actively release nucleosides upon stress, including hypoxanthine, xanthine, inosine, guanosine, and uridine[53,81–84]. Our data show that the release of xanthine and hypoxanthine occurs independently of autophagy, but instead, autophagy-dependent metabolic rewiring results in their increased intracellular accumulation. We uncover for the first time that autophagy acts as a brake on the active accumulation of nucleosides and nucleobases, which are subsequently released, either actively or passively. Release of xanthine by T cells has been identified to relay cell-extrinsic effects under stress conditions[85]. These results, together with our observations, suggest a common molecular signalling mechanism of cellular stress to the microenvironment via purine nucleobase signals. Our data further indicate that autophagy serves as a key modulator of this extracellular purine signal, which helps the cell to adapt to altered metabolic challenges such as excessive storage of fat. It is plausible that by activating nucleotide degradation, autophagy-deficient adipocytes salvage NADPH or ribose through the PPP. This enables them to partially sustain their metabolism and oxidative stress, sourcing carbon for energy, antioxidant molecules, and anabolic precursor generation. In turn, nucleoside catabolites signal the altered adipocyte state to the macrophages, which by remodelling the ECM, shut down the tissue and limit systemic dysregulation. While autophagy-dependent metabolic signals have been previously reported to play a role in cancer and inflammatory bowel disease[31,86,87], pro-fibrotic purine nucleoside catabolites have not been described before.

Yet, this study contains a few limitations warranting future investigations. First, we could not fully elucidate the exact molecular

mechanism by which purine catabolites regulate macrophage function, as we were using in vivo material only and this physiologically relevant WAT microenvironment cannot be easily recapitulated in vitro. Second, the xanthine and hypoxanthine downstream signalling pathways are poorly understood and lack an experimental toolset at the moment; however, our findings provide a strong basis for future mechanistic work when this toolset becomes available. Finally, our findings were not experimentally validated in human samples due to a lack of access to clinical materials, but we mitigated this by in silico analysis of published human datasets, strengthening the translational relevance of our findings.

In conclusion, our work highlights the key role of autophagy acting as a brake in the control of adipocyte nucleoside metabolism and tissue integrity in diet-induced obesity. When dysfunctional, this leads to uncontrolled activation of metabolic rewiring that generates purine catabolites xanthine and hypoxanthine, signalling tissue repair. By depleting autophagy, we uncover a purine nucleobase-mediated pro-fibrotic signalling pathway, and further research is necessary to elucidate whether these signalling molecules control fibrosis of other tissues and organs, potentially deeming them druggable targets.

## Methods

### Lead contact
Information and requests for reagents and resources should be directed to the Lead Contact, Anna Katharina Simon (katja.simon@kennedy.ox.ac.uk).

### Mouse models
Adipoq-Cre[ERT2] mice[88] were purchased from Charles River, UK (JAX stock number: 025124) and crossed to Atg7[fl/fl] mice[89]. Genetic deletion was induced at 6–8 weeks of age by oral gavage of 4 mg tamoxifen per mouse for five consecutive days. Tamoxifen was given to all groups of mice. Two days after receiving the last tamoxifen dose, mice were subjected to an altered diet regime with either a HFD with 60 kcal% fat (D12492i, Research Diets) or a complementary normal chow diet with 10 kcal% fat (D12450Ji, Research Diets) for the duration stated in the text. Wild-type C57BL/6J or B6.SJL.CD45.1 mice were bred in-house. Experimental cages were sex- and age-matched and balanced for genotypes. All data shown, except proteomics and metabolomics data are pooled from both sexes. Mice were maintained on a 12 h dark/light cycle and housed in groups of 3–5 with unlimited access to water and food under specific pathogen-free conditions. The temperature was kept between 20 and 24 °C, with a humidity level of 45–65%. All experiments were performed in accordance with approved procedures by the Local Review Committee and the Home Office under the project license (PPL30/3388 and P01275425).

### Bone marrow chimera generation
Recipient WT and Atg7[Ad] mice were lethally irradiated with 11 Gray dose before intravenously injecting between 250 and 300,000 B6.SJL.CD45.1 donor cells (equal numbers in the same experiment to allow comparison between the two groups). Cell replenishment was followed bi-weekly. Five weeks after irradiation, mice were treated with tamoxifen for genetic deletion and fed HFD for a total of 12 weeks (as described above).

### Tissue processing, macrophage isolation and primary cell culture
Adipose tissue digestion was performed as previously described[31]. In brief, depots were digested in DMEM containing 1% fatty acid-free BSA, 5% HEPES, 0.2 mg/ml Liberase TL (Roche), and 20 µg/mL DNaseI. Tissues were minced and incubated for 25–30 min at 37 °C at 180 rpm. Digested tissue was strained through a 300 µm mesh and the digestion was quenched by the addition of PBS with 0.5% BSA and 2 mM EDTA. Adipocyte and stromal vascular fraction were separated by 7 min centrifugation at $500 \times g$ and collected for further analysis.

To generate a conditioned medium, adipocytes were collected with wide-bore tips and washed three times with PBS. The floating fraction was collected and 250 µl of packed adipocytes were seeded in 500 µl of RPMI containing 10% foetal bovine serum (FBS) and 1% penicillin/streptomycin (P/S) and incubated for 24 h at 37 °C. Alternatively, the medium was supplemented with 10 µM forodesine. After incubation, the medium and cells were harvested, centrifuged at $300 \times g$ for 5 min and purified medium was collected and snap-frozen for in vitro experiments.

For primary cell culture, the stromal vascular fraction was enriched for CD11b[+] cells with CD11b MicroBeads (Miltenyi Biotec) according to manufacturers' instructions after red blood cell lysis. 350,000 cells were seeded in 24-well plates in RPMI containing 10% FBS and 1% P/S and incubated overnight to allow macrophages to attach. The following day, macrophages were enriched by washing the wells with room temperature PBS and treated with experimental conditions. For conditioned medium treatment, RPMI containing 10% FBS and 1% P/S was mixed with the conditioned medium in a 2:1 ratio and applied to the cells for 72 h. For nucleoside treatment, macrophages were cultured in RPMI containing 10% FBS and 1% P/S and 50 ng/ml M-CSF and treated with 100 µM of either adenosine, guanosine, hypoxanthine or xanthine over 72 h.

### Glucose tolerance test
Mice were subjected to 12 h fast before measuring fasted glucose levels. To monitor response to bolus glucose, $1.5 \times g$ of glucose per kg of body mass was injected intraperitoneally. Blood glucose levels were measured via tail clip at 15, 30, 60, 90, 120, and 180 min after injection with a glucose meter (Freestyle Lite, Abbott).

### Histology and immunohistochemistry
Adipose tissues and liver were fixed in 10% neutral buffered formalin for 24 and 48 h, respectively. All tissues were transferred to 70% ethanol and sent to the Kennedy Institute of Rheumatology Histology service for paraffin embedding, sectioning (5 µm), and staining. Haematoxylin and eosin (H&E) and picrosirius red staining were performed according to standard protocols. Images were acquired with a Zeiss Axioscan 7 scanning microscope. Image analysis and quantification were performed using QuPath, ImageJ, and an in-house developed script (available at https://github.com/Oxford-Zeiss-Centre-of-Excellence/pyHisto). In brief, the blind colour deconvolution method was used based on a stain vector estimation[90], followed by Otsu thresholding and determination of collage-to-area ratio.

For immunofluorescence staining, WAT was fixed in 4% paraformaldehyde for 24 h, embedded and sectioned as above. Tissues were subsequently deparaffinized and heat-retrieved at 100 °C for 20 min in Citrate antigen retrieval solution (Vector Laboratories, pH 6.0), and allowed to cool naturally in Tris antigen retrieval solution (10 mM, pH 8.8–9.0). Slides were washed in PBS and blocked overnight in 10% donkey serum and 3% BSA. The next day, slides were incubated with DAPI for 15 min to record a background scan. After background imaging, slides were incubated for 20 min with Fc block reagent (1:200 in 3% BSA), followed by primary antibody incubation overnight at 4 °C. Slides were washed in PBST, and incubated with secondary antibodies diluted in 3% BSA for one hour at room temperature. Following washes in PBST, slides were mounted, and images were acquired with a GE Cell DIVE multiplex imager. See the Supplementary Table 1 for a list of primary and secondary antibodies.

### Western blot
Autophagy activity was assessed by measuring autophagy flux in WAT. Adipose tissue was incubated in full DMEM supplemented with 100 nM Bafilomycin A1 and 20 mM $NH_4Cl$ in DMSO for 4 h to inhibit lysosomal activity. DMSO was used as a "vehicle" control. To assess LC3-turnover, autophagic flux was calculated as: (LC3-II (Inh) − LC3-II (Veh)). To

determine the contribution of apoptosis, WAT explants were incubated in full RPMI supplemented with either DMSO (vehicle control) or 20 μM Q-VD-OPh overnight or 10 μM staurosporine for 24 h. Protein extraction was performed as published[91]. Briefly, 500 μL of lysis buffer containing protease inhibitors and PhosphoStop was added per 100 mg of tissue. Cells were lysed using Qiagen TissueLyser II and lipid contamination was removed through serial centrifugation. Protein content was determined using a BCA Protein Assay kit and 15 μg of protein was separated on a 4–12% Bis-Tris SDS PAGE gel. After wet-transferred onto a PVDF membrane, membranes were blocked in 5 % milk in TBST, and incubated with primary antibodies overnight. Proteins were visualized on a membrane using IRDye 800 or IRDye 680 (LI-COR Biosciences) secondary antibodies at the dilution 1:10,000 (LI-COR). Quantification was performed with ImageStudio software (LI-COR). See the Supplementary Table 1 for a list of primary and secondary antibodies.

### Enzyme-linked immunosorbent assay (ELISA)
Adipose tissue secretome was generated by incubating 200 mg of adipose tissue explants (each explant ~ 10 mg) in 1 ml of RPMI containing 10% FBS and 1% P/S for 6 h at 37 °C. Cytokine levels in supernatant were measured by commercially available ELISA kits. See the Supplementary Table 1 for a list of ELISA kits. All cytokine levels were normalized to input tissue weight.

### Serum chemistry
After eight hours of fasting, serum was collected by a cardiac puncture, collected in Microtainer tubes, and centrifuged for 90 s at $15,000 \times g$. Triglycerides and high-density lipoprotein were measured using a Beckman Coulter AU680 clinical chemistry analyser.

### Gene expression analysis (qRT-PCR)
Tissues were homogenized in TRI reagent (Sigma) with ceramic beads (Bertin Instruments) using a Precellys 24 homogenizer (Bertin Instruments). RNA was extracted using RNeasy Mini or Micro Kit (Qiagen). RNA yield and quality were assessed using a NanoDrop and cDNA was synthesized using a High-Capacity RNA-to-cDNA™ kit (ThermoFisher). Gene expression was measured using TaqMan Fast Advanced Master Mix on a ViiA7 real-time PCR system. Values were normalized to *Ppia* reference gene using the comparative Ct method. Primers are listed in the Supplementary Table 1.

### Flow cytometry and cell sorting (FACS)
Cells for flow cytometry staining were isolated as described above. For surface staining, cells were incubated with fluorochrome-conjugated antibodies, LIVE/DEAD Fixable Stains and Fc receptor block antibody for 20 min at 4 °C. This was followed by a 10 min fixation with 4% PFA at room temperature. Samples were acquired on the Fortessa X-20 flow cytometer (BD Biosciences). Data were analysed with FlowJo v10.8 according to the gating strategy (Supplementary Fig. 5). See the Supplementary Table 1 for a list of flow cytometry antibodies.

### Transcriptomics (bulk RNA sequencing)
Macrophages were isolated by FACS or cultured in vitro and RNA was extracted as described above. To generate PolyA libraries, cDNA was end-repaired, A-tailed and adapter-ligated. Libraries were then size-selected, multiplexed, quality-controlled, and sequenced using a NovaSeq6000. Quality control of raw reads was performed with a pipeline readqc.py (https://github.com/cgat-developers/cgat-flow). The resulting reads were aligned to the GRCm38/Mm10 reference genome using the pseudoalignment methods Salmon[92] and Kallisto[93]. DEseq2 (v1.38.3) was used for differential gene expression analysis[94]. The workflow included the estimation of size factors, dispersion estimation, and fitting of a negative binomial generalized linear model.

Prior to differential expression analysis, batch effects attributed to sex were corrected by incorporating them within the experimental design of deseq2 analysis, using the limma package (v3.54.1) for visualisation purposes[95]. Genes were considered significantly differentially expressed based on an adjusted *p*-value threshold of <0.05, after correcting for multiple testing using the Benjamini−Hochberg procedure. To explore the biological implications of the differentially expressed genes, gene set enrichment analysis was performed using the clusterProfiler package (v4.6.0).

### Single-nucleus RNA-seq analysis
The adipocyte dataset was downloaded from the GEO database (GSE176171), originally published by Emont et al.[24]. The dataset was categorized into lean or obese groups based on BMI, following the methodology outlined in the source paper[24]. While we adopted the original study's BMI cut-offs and confounding effects, we recognise that this study has limitations in sample size, sex, age, and ethnicity distribution. To facilitate visualization, UMAP (Uniform Manifold Approximation and Projection) was recalculated, and further data analysis was conducted using the Seurat package v5.0.1[96] for single-nuclei RNA-seq analysis. Functional enrichment analysis was performed using the ClusterProfiler package (v4.6.0)[97].

### Proteomics
Adipocytes were isolated as a floating fraction upon digestion and lysed and digested in SDC buffer. Specifically, a pellet of 100 μl packed adipocytes was lysed in SDC-buffer containing 2% (w/v) sodium deoxycholate (SDC; Sigma-Aldrich), 20 mM dithiothreitol (Sigma-Aldrich), 80 mM chloroacetamide (Sigma-Aldrich), and 200 mM Tris-HCl (pH 8). After being heated at 95 °C for 10 min, the lysates were digested enzymatically using endopeptidase LysC (Wako) and sequence grade trypsin (Promega) at a protein:enzyme ratio of 50:1. The digestion process occurred overnight at 37 °C. For reversed-phase liquid chromatography coupled to mass spectrometry (LC-MS) analysis, each sample replicate was injected with 1 μg of peptide amount into an EASY-nLC 1200 system (Thermo Fisher Scientific) for separation, using a 110-min gradient. Mass spectrometric measurements were carried out using an Exploris 480 (Thermo Fisher Scientific) instrument in data-independent acquisition (DIA) mode, which utilised an isolation scheme with asymmetric isolation window sizes. The raw files were analysed using DIA-NN version 1.8.1[98] in library-free mode, with a false discovery rate cutoff of 0.01 and relaxed protein inference criteria, while employing the match-between runs option. The spectra were compared to a Uniprot mouse database (2022-03), which included isoforms. The protein intensities were normalised using MaxLFQ and filtered to ensure that each protein had at least 50% valid values across all experiments, with an additional filter to retain at least 3 valid values in at least one experimental group. The limma package[95] was used to calculate two-sample moderated *t*-statistics for significance calling. The nominal *P*-values were adjusted using the Benjamini−Hochberg method.

### Mass spectrometry
Adipocytes and conditioned medium were obtained as described above. Metabolite extractions from frozen cell pellets were performed through a two-phase extraction with 80% MeOH and chloroform. In brief, when just thawed, cells were resuspended in 500 μl ice-cold 80% MeOH, vortexed and sonicated in an ice bath for 6 min. Following 1 h of incubation on dry ice, tubes were centrifuged at $16,000 \times g$ for 10 min at 4 °C, and supernatants were mixed with water and chloroform in 1:1:1 ratio. Each sample was vortexed for 1 min and centrifuged at top speed for 15 min at 4 °C. Finally, 600 μl of the top aqueous layer was transferred to a glass vial and evaporated using an EZ-2Elite evaporator (Genevac). Samples were stored at −80 °C before analysis. The BCA assay was performed on the airdried pellets, resuspended in 200 μl of

0.2 M NaOH and heated at 95 °C for 20 min. Metabolite extractions from frozen conditioned medium precleared from cells and cell debris were performed after removing cells and cell debris by mixing 20 μl of clarified medium with 500 μl ice-cold 80% MeOH/20% $H_2O$. After vortex and 30 min incubation at −80 °C, samples were centrifuged at 16,000 × $g$ for 10 min at 4 °C, transferred to a glass vial and evaporated using an EZ-2Elite evaporator (Genevac). Dried extracts were stored at −80 °C before analysis. Dried metabolites were resuspended in 100 μL 50% ACN:water and 5 μL was loaded onto a Luna NH2 3 μm 100 A (150 × 2.0 mm) column (Phenomenex) using a Vanquish Flex UPLC (Thermo Scientific). The chromatographic separation was performed with mobile phases A (5 mM NH4AcO pH 9.9) and B (ACN) at a flow rate of 200 μl/min. A linear gradient from 15% A to 95% A over 18 min was followed by 7 min isocratic flow at 95% A and re-equilibration to 15% A. Metabolites were detected with a Thermo Scientific Q Exactive mass spectrometer run with polarity switching in full scan mode using a range of 70–975 m/z and 70.000 resolution. Maven (v 8.1.27.11) was used to quantify the targeted polar metabolites by AreaTop, using expected retention time and accurate mass measurements (< 5 ppm) for identification. Data analysis, including principal component analysis and heat map generation, was performed using in-house R scripts. In brief, metabolite intensities (area under the curve) were normalized to protein content and analysed with one-way ANOVA (ANOVA column). Metabolites with a $p$-value < 0.05 were termed as significant. Heatmaps are $z$-score normalized, assuming a normal distribution. Bar plots display relative amounts for each metabolite, calculated by averaging amounts for each condition (condition with the lowest average value set to 1). Raw metabolomics data is provided as Supplementary Data 1.

## Metabolic assays

To measure metabolite levels in cells, cells were resuspended in an ice-cold homogenization medium (0.32 M sucrose, 1 mM EDTA, and 10 mM Tris–HCl, pH 7.4) and homogenized with 72 strokes using a tight pestle. After brief sonication at 40% Amp, homogenates were centrifuged at 2000 × $g$ for 5 min at 4 °C to remove the lipid layer and protein concentration was determined using the BCA assay to normalize between conditions. 24 μg of protein was used per assay condition. Levels of xanthine and hypoxanthine (Abcam) in cell homogenates were measured using commercially available kits. Intracellular ATP was measured in living cells using the ATP bioluminescence assay kit CLS II (Roche) according to the manufacturer's instructions. Commercial kits were also used to measure xanthine and hypoxanthine (Abcam) directly in serum and conditioned medium. For more information, see the Supplementary Table 1.

## Quantification and statistical analysis

Experiments were conducted as 3 independent repeats or as indicated. Mice were randomly grouped in experimental groups and data were pooled from both sexes. Data were analysed and visualized using GraphPad Prism 9. The normal distribution of data was tested before applying parametric or nonparametric testing. For comparison between two independent groups, the unpaired Student's $t$-test or the Mann−Whitney test were applied. Comparisons across multiple groups were performed using one-way or two-way ANOVA with Dunnett, Šídák or Tukey multiple testing correction. Data were considered statistically significant when $p$-value < 0.05.

## Reporting summary

Further information on research design is available in the Nature Portfolio Reporting Summary linked to this article.

## Data availability

The RNA sequencing data generated in this study have been deposited in the GEO database under accession codes GSE263837 and GSE306424. The proteomics data generated in this study have been deposited in the PRIDE database under accession code PXD052894. The metabolomics data generated in this study are provided in Supplementary Data 1. The single-nuclei RNA sequencing data used in this study are available in the GEO database under accession code GSE176171. Source data are provided with this paper.

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

## Acknowledgements

We thank Patricia Cotta Moreira, Luke Barker, Emily Wyeth, Daniel Andrew, and Mino Medghalchi from the Biomedical Services for their responsible care and assistance with animal well-being. Histology was performed with the help of the Kennedy Institute Histology Facility, with special thanks to Dr Ida Parisi. Dr Johanna ten Hoeve–Scott at UCLA Metabolomics Center, Dr Ulrike Brüning and Dr Jennifer Kirwan at BIH Charité Berlin for their help with metabolomics sample analysis. Dr Moustafa Attar for his help with the experimental design of the transcriptomics experiment. Jonathan Webber for his help with the flow cytometry experimental design. Cell DIVE Facility team at the Kennedy Institute of Rheumatology for their help with immunofluorescent staining. This work was supported by grants from the Wellcome Trust to A.K.S. (Investigator award 220784/Z/20/Z) and M.B. (Sir Henry Wellcome Fellowship 220452/Z/20/Z), the Kennedy Trust for Rheumatology Research (KTTR) Studentship to K.P. (KEN192001) and J.K.L.K. (awarded to Marco Fritzsche), The Kenneth Rainin Foundation, and Helmholtz Association REK-0157 to A.K.S. (20220003/20230038), Clarendon Fund Scholarship to K.P., Medical Research Council Doctoral Training Partnership Grant (BRT00030) to K.P., Ramage Scholarship to K.P., PhD studentship award 203803/Z16/Z to F.C.R., Versus Arthritis grant 22617 to G.A., EPA Cephalosporin Fund to J.K.L.K., EMBO Postdoctoral Fellowship (ALTF115-2019) to A.V.L.V. Li-cor Odyssey imager was funded by ERC AdG 670930. Flow cytometry and microscopy facilities were supported by KTTR. Graphical summaries were created with BioRender.com.

## Author contributions

Conceptualization: K.P., F.P., G.A. and A.K.S. Methodology, investigation, analysis, visualization, and validation: K.P., A.H.K., F.C.R., M.B., A.V.L.-V., O.P., S.G., J.K.L.K., L.L., K.Z., M.K., H.K.H., L.K. and G.A. Essential reagents and support: O.P., L.K., P.M. and S.S. Writing of original draft: K.P., G.A. and A.K.S. Funding acquisition, supervision, and project administration: F.P., G.A. and A.K.S. Editing of the manuscript: all authors.

## Competing interests

The authors declare no competing interests.
