## [Transparent Peer Review file · Nature Communications]

Autophagy acts as a brake on obesity-related fibrosis by controlling purine nucleoside signalling

Corresponding Author: Professor Anna Katharina Simon

Version 0:

Reviewer comments:

Reviewer #1

(Remarks to the Author)

To understand the role of autophagy in high fat diet induced obesity, in this study Piletic K et al utilized an adipocyte-specific autophagy deficient genetic mouse model (inducible, adipocyte specific deletion of Atg7 gene). Findings from this study provides a novel understanding on the role of autophagy in diet induced obesity in mouse. HFD induced obesity upregulates autophagy (at least at the early stage). However, deficiency of autophagy in HFD condition may lead to uncontrolled metabolic rewiring of adipocytes in mouse, specifically excessive purine nucleoside catabolism leads to release of Xanthine and Hypoxanthine metabolites. Interestingly, the data in this paper suggested that these metabolites released from adipocytes in autophagy-deficient HFD-induced obesity condition initiates an adipocyte-macrophage cross-talk signaling and triggers a switch of macrophage phenotype from pro-inflammatory NPVM (Lyve1High- MHCII low) to tissue-reparative pro-fibrotic PVM (Lyve1 low -MHCII high) macrophages, cause transcriptional induction of extracellular matrix genes, and leads to higher fibrosis in adipose tissue. Identification of metabolites- Xanthine and Hypoxanthine as profibrotic signal is a novel finding of this study. Several methodological aspects of this study, and interpretation of publicly available human data is key to this study but is unclearly presented and need further clarification.

1) The first piece of data presented in this paper utilized publicly available human adipose sn-RNAseq dataset (Emont et al, 2022, Nature) to identify, and provide evidence in support of the role of autophagy in Human (Figure 1A and 1B and Figure 2G and H). Authors used Seurat package to define Differentially expressed genes (DEGs) between lean and obese and identify enriched pathways. This public snRNA-seq data includes "paired subcutaneous (SAT) and visceral (VAT) adipose tissue from 10 individuals, and SAT alone from three additional individuals (10 women, 3 men)" of these individuals 4 with BMI <30 kg/m² and 6 individual with BMI >40 kg/m². This sample size is insufficient for any robust inference on obesity. Also, this BMI cutoff is not complaint with standard guidelines of obesity. The cohort also included bot sex (males and female), Caucasian and African ancestry. The statistical analysis strategy for identification of DEG in this dataset is not described appropriately. How different tissue (SAT and VAT), sex, age, ethnicity was taken into consideration in analysis? instead of simple Wilcoxon Rank Sum test, whether Pseudo bulk analysis strategy or General linear mixed model/two-part hurdle model-MAST was used to define DEG after controlling for confounding factors and prevent inflation of error rate, is not clear.

2) It appears that the inducible, adipocyte-specific deletion of Atg7 mouse model was developed as a part of previously published study by this group investigating gut-adipocyte interaction/crosstalk (Richter FC, EMBOJ, 2023). Rationale for selecting Atg7, but not other autophagy regulatory genes need to be mentioned.

3) An important weakness of the manuscript is not being able to rationalize the use the methods before presenting the result and interpretation. Despite the journal specified format of the paper (methods described at the end) please connect the choice of method with result. Reference for key methods for this paper, for example "autophagy flux" analysis using western blot is not described. Define "autophagy flux" and why this method was chosen, can it be considered as a "gold standard"? What are the limitations?

4) In Human histological and multiomic studies ECM proliferation and adipose tissue fibrosis in obesity is shown to strongly associated with detrimental metabolic consequences of obesity. Mouse model with knockout of ECM components is shown to be metabolically heathy despite genetic and diet induced obesity background. Surprisingly, in autophagy deficient adipocyte-specific deletion of Atg7 mouse model in 12 to 16-week HFD showed high fibrosis but some features of metabolic heath including lower glucose AUC, lower liver TG (Figure S4). However, serum and adipose adiponectin level was lower. Author suggests a theory of induction of profibrotic macrophage, difference in gWAT and iWAT in systemic impact etc. How this finding matches with published studies on human obesity? Was their evidence that individuals with lower autophagy is metabolically heathy and have macrophage characteristics, cytokine profiles etc. like mice model described here. Is there

genetic evidence in human suggesting that lower Atg7 is associated with metabolic health despite obesity. Relevance of this study to human obesity is lacking.

- 5) Large number of metabolomic data is available from human cohorts. Is their evidence about upregulation of serum purine metabolites like xanthine and hypoxanthine was observed in human obesity or in a subset of metabolically healthy obesity?
- 6) An important novel finding of this study is autophagy deficient adipocyte derived metabolites- Xanthine and Hypoxanthine mediated profibrotic signal triggering a switch of macrophage phenotype. How is this signal pathway triggered? In expansion of Figure 6D and 6E in vitro experiments, author should generate a transcriptomic and proteomic profile to understand Xanthine and Hypoxanthine mediated signaling in macrophages (ATMs) to elucidate this putatively novel signaling pathway.
- 7) No table presented as "key resources table", but supplementary table-3 is presenting the resources, please refer appropriate table number.
- 8) Proteomics data accession is marked as XXXX, please include a usable accession.
- 9) Manuscript Title should reflect that this study is done in Mouse model (not validated in human obesity).
- 10) Discuss limitations of this study.

Reviewer #2

(Remarks to the Author)

In the manuscript by Piletic et al, the authors show that gonadal adipocytes from HFD-induced obese mice have increased autophagic flux, and blockade of autophagy induces fibrosis. The authors conclude that a blockade in autophagy leads to elevated secretion of xanthine and hypoxanthine from obese adipocytes, which recruits/polarizes adipose macrophages into tissue-reparative and fibrosis-inducing macrophages. The data presented in the manuscript are solid and thorough and are strengthened by the associations with human adipose tissue data analysis. However, there are a few issues that require additional data that should be addressed to better test hypotheses raised by the data and to strengthen the conclusions of the manuscript:

1. At 9 weeks of HFD feeding Atg7KO mice have a significant increase in fibrosis (Fig 1E), but this precedes the elevation of autophagic flux observed in HFD adipose tissue (Fig 1C). Can the authors please comment on this discrepancy? Is the Atg7KO-induced fibrosis independent of HFD-feeding? Importantly, are xanthine or hypoxanthine levels elevated at this 9-week timepoint?
2. In Figure 1C, it's shown that autophagy flux is decreased at 60 weeks of HFD. Does this timepoint correlate with the amount of fibrosis in the adipose tissue; i.e. does fibrosis increase when autophagy decreases?
3. The authors present interesting data that show despite increased gWAT fibrosis, glucose homeostasis is significantly improved in Atg7KO mice, perhaps due to improved fat storage in the subcutaneous depot. It's been shown that macrophages in this depot also significantly affect adipose tissue function. How does Atg7KO affect the subcutaneous depot with regard to xanthine/hypoxanthine secretion and macrophage phenotypes?
4. Lastly, in Figure 3G, it's shown that overnight treatment of adipose explants with the apoptosis inhibitor does not affect secretion of xanthine/hypoxanthine. Overnight treatment is not a long time frame, especially in the context of 16 weeks of HFD feeding, thus the effect of this treatment could be underestimated. As a counter experiment, it would be useful to see if inducing apoptosis can increase xanthine/hypoxanthine secretion in explants to rule out apoptosis as a contributing factor to the xanthine secretion observed in Atg7KO adipocytes.
5. Minor comment: could the authors comment on how a blockade in autophagy could lead to increased secretion of these metabolites?

Reviewer #3

(Remarks to the Author)

In this study Piletic et al. report that loss of Atg7 (an autophagy effector) in white adipose tissue leads to 1) fibrosis, 2) macrophage infiltration, 3) release of purine catabolites from adipocytes in the setting of high-fat diet. The authors utilize conditional mouse model for Atg7ko, tissue explants, cultured primary adipocytes and conditioned medium from the adipocytes. Links between autophagy and fibrosis and macrophages are known. The novelty of the manuscript is the link between adipose autophagy and purine metabolism, which is also the facet of the manuscript that the authors could have investigated further.

Specific comments

- 1) Why does hypoxanthine/xanthine accumulate intracellularly, and why are they secreted more when autophagy is blocked in adipocytes? What is the underlying mechanism. This is a key missing point in the manuscript.
- 2) Perilipin1 staining is not a marker of adipocyte viability. The 'perilipin1 negative adipocytes' in Fig. S2C,D are presumably non-adipocytes (macrophages and other the immune cells) in the tissue. It is not clear how the authors conclude that perilipin1-negative adipocytes are in fact adipocytes. This is contradictory to what is shown in Fig. 4E wherein adipocyte numbers between the two groups remain similar.
- 3) Many of the experiments rely on isolated adipocytes from wt and Atg7ko adipose tissue which have high cell death (Fig. S2E). Are the phenotypes of high hypoxanthine/xanthine release, macrophage infiltration, lower adipocyte markers all a result of cell death?
- 4) Do intracellular levels of xanthine/hypoxanthine change in Atg7ko obese adipocytes?
- 5) Are the pathways shown in Fig. 1B upregulated or downregulated in obese state?
- 6) Reporting: The manuscript could use a considerable effort in reporting the n values and putting more details in legends

and data shown. The reporting summary says that the exact n is reported; however, the authors are urged to revisit each figure panel and be consistent in the reporting of sample size. To cite a few instances - What is the n for Fig. S2E? The data shows 4 points. The legend mentions 5-6 mice. Is this a typo? Or are all data not being reported? Similarly, what is the n for Fig. S2D? The legend mentions 7-8 mice, and the graph shows data points that are >50. What are these data points? Fields? The authors also show data from blots without showing the blots themselves.

Other comments

1) There is robust decrease in Atg7 in the Atg7cko (Fig. 1A). Despite this, there is considerable signal of LC3II in the Atg7cko samples (Fig. S1B). Perhaps this is due to WT and Atg7ko samples being run on separate gels. The flux calculations are valid. Separate blots make it seem that LC3II levels are similar between WT and Atg7ko mice on HFD which is most likely not the case.

2) Is the KO specific to adipocytes? What is the status of Atg7 in other tissues?

Version 1:

Reviewer comments:

Reviewer #1

(Remarks to the Author)

in this revised manuscript reviewed earlier by me, authors carefully considered all of the reviewers concern, and addressed them with additional experimental data, edited the manuscript text and figures thoroughly, included limitations of the study and relevance to human, added appropriate supplementary table details (please correct the supplementary resources table with proteomics data id). It is an important work, and wish that authors peruse this work more on human.

Reviewer #2

(Remarks to the Author)

The revised manuscript is much improved and addresses the points raised in my review.

Reviewer #3

(Remarks to the Author)

Remaining comments:

Regarding the use of triton-X-100 to distinguish total intracellular metabolite pool, the term "secretion" should be revised to "release" or "lysis-induced release" in all places - to avoid implying an active export mechanism where none is demonstrated. The authors say as much but only in some places. Please correct.

Regarding Fig. S1C, reporting samples on separate blots sets a wrong precedent for future readers. The limitation is not methodology of western blotting in adipose tissue, but your inertia of rest in not running all samples in one blot as previously requested. Stating "Limited increase in LC3B-II is likely influenced by tissue heterogeneity and technical limitations of adipose tissue western blotting." is erroneous because the limitation is the fact that they are on separate blots and not what is stated. All the other blots reported evidently do not have those limitations.

REVIEWER COMMENTS

Reviewer #1 (Remarks to the Author):

To understand the role of autophagy in high fat diet induced obesity, in this study Piletic K et al utilized an adipocyte-specific autophagy deficient genetic mouse model (inducible, adipocyte specific deletion of Atg7 gene). Findings from this study provides a novel understanding on the role of autophagy in diet induced obesity in mouse. HFD induced obesity upregulates autophagy (at least at the early stage). However, deficiency of autophagy in HFD condition may lead to uncontrolled metabolic rewiring of adipocytes in mouse, specifically excessive purine nucleoside catabolism leads to release of Xanthine and Hypoxanthine metabolites. Interestingly, the data in this paper suggested that these metabolites released from adipocytes in autophagy-deficient HFD-induced obesity condition initiates an adipocyte-macrophage cross-talk signaling and triggers a switch of macrophage phenotype from pro-inflammatory NPVM (Lyve1High- MHCII low) to tissue-reparative pro-fibrotic PVM (Lyve1 low -MHCII high) macrophages, cause transcriptional induction of extracellular matrix genes, and leads to higher fibrosis in adipose tissue. Identification of metabolites- Xanthine and Hypoxanthine as profibrotic signal is a novel finding of this study. Several methodological aspects of this study, and interpretation of publicly available human data is key to this study but is unclearly presented and need further clarification.

1) The first piece of data presented in this paper utilized publicly available human adipose sn-RNAseq dataset (Emont et al, 2022, Nature) to identify, and provide evidence in support of the role of autophagy in Human (Figure 1A and 1B and Figure2G and H). Authors used Seurat package to define Differentially expressed genes (DEGs) between lean and obese and identify enriched pathways. This public snRNA-seq data includes “paired subcutaneous (SAT) and visceral (VAT) adipose tissue from 10 individuals, and SAT alone from three additional individuals (10 women, 3 men)” of these individuals 4 with BMI <30 kg/m² and 6 individual with BMI >40 kg/m². This sample size is insufficient for any robust inference on obesity.

We acknowledge the reviewer's observation regarding the sample size of the publicly available dataset (Emont et al., 2022). This study is the first to explore adipocytes by snRNA-seq in adipose

tissue from lean and obese individuals. Despite its limitations, this dataset was rigorously peer-reviewed and has been highly cited and used in different studies to enable research translatability, underscoring its scientific merit and relevance to the field. In this study, we utilized this dataset to explore whether autophagy-related pathways exhibit alterations in human adipose tissue. The fact that we observed significant changes in autophagy-related pathways, despite the relatively small sample size, suggests that these findings are valuable and that further investigations are warranted. By leveraging this dataset and providing further validation in mice, we wanted to pave the way for future studies in humans to build upon our findings with expanded sample sizes. We have now included these limitations to the single nucleus RNA-seq analysis in the Materials and Methods section, page 26 as: ***“While we adopted the original study’s BMI cutoffs and confounding effects, we recognise that this study had limitations in sample size, sex, age, and ethnicity distribution.”***

Also, this BMI cutoff is not compliant with standard guidelines of obesity.

To ensure consistency and comparability, we chose to use the same terminology as the original study (Emont et al., 2022). According to WHO guidelines, individuals with BMI ≤ 25 kg/m² are categorized as "lean" while those with BMI > 40 kg/m² are classified as "severely obese". Their figures, however, refer to these groups using BMI cut-offs of < 30 and > 40 . As our re-analysis built on their published dataset, we maintained the original cut-off to facilitate direct comparisons and avoid inconsistencies in interpretation.

The cohort also included both sex (males and female), Caucasian and African ancestry. The statistical analysis strategy for identification of DEG in this dataset is not described appropriately. How different tissue (SAT and VAT), sex, age, ethnicity was taken into consideration in analysis?

Our analysis is based on the processed Seurat object from Emont et al. (2022), who addressed demographic variables like sex and age in the initial data processing. We conducted a focused re-analysis, and thus our study is limited to the scope and demographic representation provided in the original dataset. Furthermore, given the small sample size, accounting for multiple cofactors would not yield meaningful results. We recognize that potential confounding effects from these factors are best accounted for during initial data collection and processing, as detailed in the original publication. As mentioned already above, we have now included these limitations to description of the single nucleus RNA-seq analysis in the Materials and Methods section, page 26 as: ***“While we adopted the original study’s BMI cutoffs and confounding effects, we recognise that this study has limitations in sample size, sex, age, and ethnicity distribution.”***

Instead of simple Wilcoxon Rank Sum test, whether Pseudo bulk analysis strategy or General linear mixed model/two-part hurdle model-MAST was used to define DEG after controlling for confounding factors and prevent inflation of error rate, is not clear.

We thank the reviewer for pointing out the need to clarify our differential expression strategy. In the original analysis, we used the Wilcoxon Rank Sum test, a robust, non-parametric approach commonly applied in single-cell RNA-seq studies to identify DEGs between lean and obese adipocytes. To address concerns about inflation of type-I error and the need to control for covariates, we re-ran our human snRNA-seq data through a two-part hurdle model using MAST.

When we applied the MAST hurdle model, we identified 1,999 DEGs (adjusted p-value < 0.05), compared to 1,793 DEGs from the Wilcoxon test. Notably, 1,791 genes overlapped between the two

methods (90% concordance), and all GO terms originally flagged as significant (e.g., “macroautophagy”, “response to insulin”, “adaptive thermogenesis”) remained under MAST DEG. Nevertheless, we observed modestly different p-adjusted values, as expected when switching statistical frameworks (Reviewer Figure 1A–C). This high overlap indicates that our core biological conclusions (autophagy pathway alterations in obese human adipocytes) are not artefacts of a specific test, but rather reflect a true transcriptional signature. Therefore, while we recognise that pseudo-bulk methods and mixed models offer alternatives, we prioritised methodological consistency with prior studies, enabling clearer comparative insights.

Reviewer Figure 1: Comparison of the Wilcoxon Rank Sum test and two-part hurdle model-MAST for analysis of human adipose tissue sc-RNAseq data. A) Overlap of differentially expressed genes (DEGs) identified by two different statistical methods. B-C) Enrichment GO analysis of differentially regulated pathways in human adipocytes isolated from obese compared to lean WAT. Figure provided in the original manuscript (B, corresponds to Fig. 1B), where DEGs were identified by the Wilcoxon Rank Sum test. Re-analysed figure where DEGs were identified by a two-part hurdle model-MAST (C).

2) It appears that the inducible, adipocyte-specific deletion of Atg7 mouse model was developed as a part of previously published study by this group investigating gut-adipocyte interaction/crosstalk (Richter FC, EMBOJ, 2023). Rationale for selecting Atg7, but not other autophagy regulatory genes need to be mentioned.

We thank the reviewer for asking to provide a rationale for selecting Atg7. We selected Atg7 as the target gene based on its critical role in the regulation of autophagy. This has been added to the manuscript on page 4, paragraph 2: ***“Specificity of Cre expression in adipose tissues and Atg7 expression in metabolic tissues were assessed prior to this study (Richter et al., 2023). Atg7 encodes for an E1-like enzyme required for autophagosome formation, and its deletion leads to a profound loss of autophagic activity (Dikic and Elazar, 2018). Through its ligase activity, Atg7 is essential for the conjugation of LC3 to autophagic membranes, which is a critical step in the autophagy pathway (Levine and Klionsky, 2004).”*** Due to this, Atg7 has been widely used to study autophagy deficiency in various tissues and conditions, as its deletion results in a profound ablation of autophagic activity (Mortensen et al., 2011, Zhou et al., 2022, Richter et al., 2023, Cai et al., 2018, Sakane et al., 2021).

3) An important weakness of the manuscript is not being able to rationalize the use the methods before presenting the result and interpretation. Despite the journal specified format of the paper (methods described at the end) please connect the choice of method with result.

We thank the reviewer for this observation. In the original manuscript, we aimed to maintain clarity and conciseness by linking the precise methods to the results mostly within the figure legends. This ensured that the rationale for the choice of method was presented in context and adherence to the journal's specified format of the paper. Nevertheless, we have now expanded the results with the choice of methods and are highlighting them here:

- ***“Following the induction of Atg7 deletion in mature adipocytes (Fig S1B), the obesity-associated increase in autophagy flux was abrogated in gonadal WAT (gWAT) adipose depot after the tamoxifen treatment, as assessed by western blot analysis at 16 weeks after HFD treatment (Fig S1C-D).”*** (page 4, paragraph 2)
- ***“In line with the aggravated ECM accumulation, qPCR analysis revealed that several ECM components and enzymes, including Col3a1, Fn1, Mmp14, and Timp1, were strongly increased (Fig 1G).”*** (page 5, paragraph 1)
- ***“In addition, obese Atg7Ad mice displayed no significant changes in circulating levels of adiponectin or leptin, as measured by ELISA, compared to controls (Fig S4J-K).”*** (page 8, paragraph 2)
- ***“To assess the production and release of these cytokines, we cultured gWAT ex vivo for six hours and measured their secretion by ELISA.”*** (page 8, paragraph 3)
- ***“Flow cytometry analysis also demonstrated that the endothelial cell population (CD31+) remained steady (Fig 4H).”*** (page 9, paragraph 1)

Reference for key methods for this paper, for example “autophagy flux” analysis using western blot is not described. Define “autophagy flux” and why this method was chosen, can it be considered as a “gold standard”? What are the limitations?

One of the main challenges in measuring autophagy is interpreting increased formation versus impaired degradation. Instead of just using LC3B as a marker for autophagy increase or decrease, it is essential to assess autophagy flux. Autophagy flux takes into account both the formation of autophagosomes (as indicated by LC3B) and their degradation (including LC3B) by lysosomes. To

accurately measure autophagy induction or reduction, it is crucial to inhibit lysosomal activity. We have carefully validated this approach in previous studies working with primary cells and tissues over the past 10 years, both within our team and others (Richter et al., 2023, Alsaleh et al., 2020, Morselli et al., 2010). Western blot with lysosomal inhibitors is the gold standard in autophagy research. In this paper, and according to the guidelines for the use and interpretation of assays for monitoring autophagy (Klionsky et al., 2021), we used western blot-based analysis of LC3B-II turnover in the presence and absence of lysosomal inhibitors (cocktail of bafilomycin A1 and ammonium chloride), a widely accepted and reliable method to measure autophagy flux. We carefully optimized the concentrations of this drug cocktail and the timing of treatment for adipose tissue. This approach is now explained in more detail on page 24 as ***“Autophagy activity was assessed by measuring autophagy flux in WAT. Adipose tissue was incubated in full DMEM supplemented with 100 nM Bafilomycin A1 and 20 mM NH4Cl in DMSO for 4 h to inhibit lysosomal activity. DMSO was used as a ‘vehicle’ control. To assess LC3-turnover, autophagic flux was calculated as: (LC3-II (Inh) – LC3-II (Veh)).”***

4) In Human histological and multiomic studies ECM proliferation and adipose tissue fibrosis in obesity is shown to strongly associated with detrimental metabolic consequences of obesity. Mouse model with knockout of ECM components is shown to be metabolically healthy despite genetic and diet induced obesity background. Surprisingly, in autophagy deficient adipocyte-specific deletion of Atg7 mouse model in 12 to 16-week HFD showed high fibrosis but some features of metabolic health including lower glucose AUC, lower liver TG (Figure S4). However, serum and adipose adiponectin level was lower. Author suggests a theory of induction of profibrotic macrophage, difference in gWAT and iWAT in systemic impact etc. How this finding matches with published studies on human obesity? Was their evidence that individuals with lower autophagy is metabolically healthy and have macrophage characteristics, cytokine profiles etc. like mice model described here. Is there genetic evidence in human suggesting that lower Atg7 is associated with metabolic health despite obesity. Relevance of this study to human obesity is lacking.

We thank the reviewer for raising an important topic that was not broadly discussed in the original manuscript. It is widely reported that autophagy dysregulation is closely associated with metabolic health outcomes in humans. These appear to be organ/tissue-dependent, therefore, both induction or suppression of autophagy has been linked to metabolic health and obesity (Zhang et al., 2018). Thus, we would like to cite three studies that highlight the relevance of this study to human obesity, a shorter version is now part of the manuscript.

Autophagy deletion may serve as an experimental model to study aging since autophagy activity drastically declines with age across diverse species and organs (Zhang et al., 2022, Amorim et al., 2022, Kitada and Koya, 2021). Age is also an important risk factor for obesity, driving the decline of metabolic health (Kuk et al., 2009, Ou et al., 2022). It was recently reported that in humans, caloric restriction prevents macrophage activation and, consequently, fibrosis (Yu et al., 2024). While the authors have not studied autophagy, it is plausible that autophagy, at least to some extent, contributes to the observations they made, considering that caloric restriction is one of the most potent non-genetic inducers of autophagy (Bagherniya et al., 2018).

Another example is a human biallelic MFN2 (mitofusin 2, for mitochondrial fusion) mutation, which leads to upper body adipose hyperplasia and reduction of both leptin and adiponectin secretion

from WAT (Rocha et al., 2017). Mfn2-deficient adipocytes display mitochondrial dysfunction-induced cellular stress and impaired autophagy - perhaps exhausted mitophagy (Sebastián et al., 2016), in line with our observations of altered fat distribution, decrease in adipokine secretion and mitochondrial protein expression in Atg7-deficient adipocytes. As we highlighted in our study, the authors faced similar difficulties disentangling the contribution of upper body fat vs lower limb adipose depots to metabolic health due to possible depot-specific contribution (Rocha et al, 2017).

Furthermore, IRF5-deficient mice (IRF-5 is a transcription factor directing pro-inflammatory macrophage phenotype) on a high-fat diet exhibited increased anti-inflammatory macrophage polarisation, leading to collagen deposition and fibrosis in gWAT alongside expansion of iWAT. Notably, these mice showed enhanced insulin sensitivity despite the fibrotic changes. Importantly, in obese human subjects, IRF5 expression in visceral adipose tissue was negatively correlated with insulin sensitivity and collagen deposition, linking fibrosis with metabolic health in humans. This study suggests that specific macrophage polarisation and fibrotic remodelling in adipose tissue can contribute to a metabolically healthier obesity phenotype, resonating with our findings of profibrotic macrophage induction and depot-specific effects in Atg7-deficient mice (Dalmas et al., 2015).

Finally, in genome-wide association studies (GWAS), Atg7 has been strongly linked to body fat distribution (Dornbos et al., 2022, <https://t2d.hugeamp.org/gene.html?gene=ATG7>), suggesting either an autophagy-dependent or a non-canonical role of Atg7. While the available data do not provide any information on either a positive or negative association of Atg7 with the fat distribution, they nevertheless imply a role for Atg7 in humans, aligning with our observations in mice.

We have extended the discussion of the original manuscript to comment on the relevance to human obesity on pages 13-14 as: ***“The role of autophagy in adiposity remodelling has also been observed in humans, where genome-wide association studies strongly linked Atg7 to body fat distribution (Dornbos, 2022 #652}. In addition, autophagy impairment, as seen in patients with biallelic MFN2 mutation (mitofusin 2, involved in mitochondrial fusion), was similarly found to impact depot-specific fat expansion {Rocha, 2017 #653;Sebastian, 2016 #654}. Thus, a better understanding of autophagy as a determinant of WAT remodelling and fibrosis holds important therapeutic potential to improve obesity management and health outcomes of obese patients.”*** and ***“In addition, caloric restriction studies in humans, known to induce autophagy, demonstrated reduced macrophage activation and, consequently, protection from adipose tissue fibrosis but the link to autophagy was not made {Yu, 2024 #655}.”***

5) Large number of metabolomic data is available from human cohorts. Is their evidence about upregulation of serum purine metabolites like xanthine and hypoxanthine was observed in human obesity or in a subset of metabolically healthy obesity?

Indeed, there are several lines of evidence showing that serum purine metabolites are upregulated in human obesity which were also cited in the original manuscript page 14: ***“Notably, we found xanthine and hypoxanthine increased with obesity progression in mouse serum, and similar***

observations have been made before in humans, identifying adipose tissue as one of the main contributing factors (Nagao et al., 2018, Furuhashi et al., 2020, Ho et al., 2016, Xie et al., 2014)."

6) An important novel finding of this study is autophagy deficient adipocyte derived metabolites- Xanthine and Hypoxanthine mediated profibrotic signal triggering a switch of macrophage phenotype. How is this signal pathway triggered? In expansion of Figure 6D and 6E in vitro experiments, author should generate a transcriptomic and proteomic profile to understand Xanthine and Hypoxanthine mediated signaling in macrophages (ATMs) to elucidate this putatively novel signaling pathway.

As described in the original manuscript, we have found that autophagy-deficient adipocytes signal through increased release of xanthine and hypoxanthine, which induces a tissue-reparative phenotypic switch in macrophages. Specifically, we observed that xanthine, and to a lesser extent hypoxanthine, but not adenosine and guanosine, significantly upregulated ECM-related genes in adipose tissue macrophages *in vitro*. We were unable to pursue a proteomics approach in the current study due to too limited ex vivo material. In response to the reviewer's request, we have now generated a transcriptomic profile of adipose tissue macrophages, which were isolated from lean visWAT, cultivated in the presence of 50 ng/ml M-CSF and treated with 100 μ M xanthine or hypoxanthine in full RPMI *in vitro* over 72h. RNA was isolated with an RNAeasy micro plus kit and bulk RNAseq was performed.

Gene ontology (GO) enrichment analysis of our transcriptomic data highlighted significant upregulation of MAPK and ERK signalling pathways in response to xanthine and hypoxanthine treatment. Additionally, we observed enrichment in GTPase activity and Ras signalling, further supporting the activation of these intracellular pathways. KEGG Reactome pathway analysis reinforced these findings, demonstrating significant upregulation of Rho, Rac, and MAPK pathways upon stimulation. Notably, Rho signalling has been implicated in promoting ERK translocation to the nucleus, facilitating downstream transcriptional activation (Liu et al., 2004). Importantly, the MAPK-ERK cascade has been widely reported to be activated by GPCRs, including purinergic receptors (Sugden and Clerk, 1997). ERK activation plays a crucial role in cell growth, proliferation, and fibrosis (Sun et al., 2015, Wen et al., 2022). In macrophages, ERK activation is associated with an anti-inflammatory reparatory state, which promotes tissue repair (Shirakawa et al., 2020).

Furthermore, previous studies have demonstrated that purine species can induce RhoA activation via P2Y₁₀, supporting a mechanistic link between purinergic signalling and Rho-mediated ERK activation (Gurusamy et al., 2021). While direct stimulation of P2Y₁₀ by xanthine or hypoxanthine remains speculative in our settings, our findings align with existing literature suggesting that these purines may stimulate P2Y₁₀ to trigger the MAPK-ERK signalling cascade, ultimately driving a pro-repair macrophage phenotype. Our analysis identified P2Y₁₀ as one of the most upregulated genes following stimulation with xanthine and hypoxanthine. While P2Y₁₀ is classified as a purinergic receptor, it remains officially designated as an orphan GPCR by the International Union of Basic & Clinical Pharmacology (<https://www.guidetopharmacology.org>). Nevertheless, previous studies have suggested that P2Y₁₀ can couple to the G_{12/13} family of G-proteins, modulating intracellular signalling cascades, including cAMP levels and downstream kinase activation (Shinjo et al., 2017, Yin et al., 2024). Notably, recent work has demonstrated that rewiring of intracellular nucleotide

metabolism can influence macrophage functional states (John et al., 2023), underscoring the broader relevance of purine metabolism in immune cell reprogramming. While our findings focus on extracellular purine-mediated signaling, they complement this emerging paradigm by suggesting that purine metabolism may act as both an extracellular and intracellular regulator of macrophage phenotype. Nevertheless, we believe that further experimental characterization falls beyond the scope of the current study.

We included these new data as Fig 6F-G along with the following comment in the manuscript in the result section on page 11: ***“To shed light on signalling pathways induced by xanthine and hypoxanthine in macrophages, we performed transcriptomic profiling of adipose tissue macrophages treated with xanthine or hypoxanthine in vitro. This revealed significant upregulation of MAPK and ERK signalling pathways, alongside enrichment in GTPase activity and Ras signalling transcripts (Fig 6F-G). These pathways are commonly triggered by G-protein coupled receptors (GPCRs), including purinergic receptors {Sugden, 1997 #656}. ERK activation is associated with a reparative macrophage phenotype and fibrosis {Shirakawa, 2020 #657}{Sun, 2015 #659}{Wen, 2022 #660}. Notably, P2Y10, a purinergic GPCR, and one of the most upregulated genes following xanthine/hypoxanthine treatment in our data, has been linked to RhoA activation and downstream ERK signalling {Gurusamy, 2021 #658}. These findings suggest that extracellular purines may reprogram macrophages via purinergic GPCR-mediated MAPK-ERK signalling.”*** and in the discussion on page 14: ***“Our data suggest that this effect may be mediated via purinergic GPCR signalling, particularly through P2Y10, leading to activation of the MAPK-ERK pathway and downstream profibrotic transcriptional programs.”***

F)

G)

NEW Figure 6F-G: Transcriptomics analysis of macrophages isolated from lean WT gWAT and cultivated in vitro in the presence of 50 ng/ml M-CSF and treated with 100 μ M xanthine or hypoxanthine for 72 hours. Hypoxanthine and xanthine-treated conditions were compared to the control, and shared DEGs were analysed. F) Enrichment gene ontology (GO) analysis of shared upregulated pathways in macrophages treated with either xanthine or hypoxanthine compared to the control. The number of genes identified for each term is labelled. G) Dot plot of scaled expression of key DEGs regulating GTPase-mediated signal transduction, MAPK signalling pathway and tissue remodelling. $n = 3-4$ mice.

7) No table presented as “key resources table”, but supplementary table-3 is presenting the resources, please refer appropriate table number.

We thank the reviewer for pointing this out. We have now updated the reference to the table of key resources used in this study, which is now labeled as Supplementary Table 3.

8) Proteomics data accession is marked as XXXX, please include a usable accession.

We thank the reviewer for pointing this out. We have now updated the proteomics data accession reference: PXD052894, Token: 01IOTjSdpMyF.

9) Manuscript Title should reflect that this study is done in Mouse model (not validated in human obesity).

Thank you for your suggestion. While we have chosen not to modify the manuscript title in order to maintain its conciseness, we agree that it is important to clarify the study context. We have revised the abstract to explicitly state that this study was conducted in a mouse model and to highlight the relevance of these findings to human obesity. This should help provide clearer context regarding the preclinical model used and its implications for human studies. The following alterations were made to the abstract: ***“Using a mouse model, we demonstrate that autophagy is a key tissue-specific regulator of WAT remodelling in diet-induced obesity.”*** and ***“Our findings in mice reveal a novel role for adipocyte autophagy in regulating tissue purine nucleoside metabolism, thereby limiting obesity-associated fibrosis and maintaining the functional integrity of visceral WAT.”***

10) Discuss limitations of this study.

We have included a paragraph in the discussion on page 15 to highlight limitations of this study as follows: ***“Yet, this study contains a few limitations warranting future investigations. First, we could not fully elucidate the exact molecular mechanism by which purine catabolites regulate macrophage function as we were using in vivo material only and this physiologically relevant WAT microenvironment cannot be easily recapitulated in vitro. Second, the xanthine and hypoxanthine downstream signalling pathways are poorly understood and lack an experimental toolset at the moment, however, our novel findings provide a strong basis for future mechanistic work when this toolset becomes available. Finally, our findings were not experimentally validated in human samples due to a lack of access to clinical materials, but we mitigated this by in silico analysis of published human datasets, strengthening the translational relevance of our findings.”***

Reviewer #2 (Remarks to the Author):

In the manuscript by Piletic et al, the authors show that gonadal adipocytes from HFD-induced obese mice have increased autophagic flux, and blockade of autophagy induces fibrosis. The authors

conclude that a blockade in autophagy leads to elevated secretion of xanthine and hypoxanthine from obese adipocytes, which recruits/polarizes adipose macrophages into tissue-reparative and fibrosis-inducing macrophages. The data presented in the manuscript are solid and thorough and are strengthened by the associations with human adipose tissue data analysis. However, there are a few issues that require additional data that should be addressed to better test hypotheses raised by the data and to strengthen the conclusions of the manuscript:

1. At 9 weeks of HFD feeding Atg7KO mice have a significant increase in fibrosis (Fig 1E), but this precedes the elevation of autophagic flux observed in HFD adipose tissue (Fig 1C). Can the authors please comment on this discrepancy?

As the reviewer correctly points out, the increase in fibrosis after 9 weeks of HFD feeding in Atg7Ad mice precedes the elevation of autophagic flux in WT mice. We are not surprised by this discrepancy, considering that 9 weeks of HFD refers to early-to-mid obesity (Lin et al., 2000). While many of the gWAT changes we observe in our model, including the fibrotic process, are a common consequence of obesity, the entire progression of the disease is accelerated in our Atg7Ad model. Our data suggest that under autophagy replete conditions, the extended high-fat diet leads eventually to increased cellular stress in adipocytes requiring its increased use of autophagy. If we deplete autophagy in these settings, we block the adipocyte's ability to alleviate obesity-induced stress. Therefore, in Atg7Ad mice, adipocytes are already prone to higher levels of cellular stress thus inducing tissue fibrosis earlier (around 9 weeks post-HFD). This was added to the manuscript on page 13 as: ***“While chronic fibrosis occurs in gWAT of HFD-fed wild-type mice progressively, this fibrotic development was accelerated in adipose tissue deficient for adipocyte autophagy. The early fibrotic onset observed in Atg7Ad gWAT, preceding elevation of autophagy flux in WT mice, suggests there is a demand for autophagy to maintain adipocyte cellular homeostasis early on during dietary challenge. This requirement for adipocyte autophagy is further exacerbated with the prolonged HFD feeding, likely to reduce diet-induced cellular stress.”***

Is the Atg7KO-induced fibrosis independent of HFD-feeding?

Our results show that Atg7Ad-induced fibrosis is dependent on HFD feeding, as mice fed NCD do not display any difference in collagen accumulation after 16 weeks of altered diet (new Supplementary Figure 1E-F). This is unsurprising considering previous reports, which demonstrate that fibrosis development is induced by obesity (Marcelin et al., 2022, Sakers et al., 2022, Gliniak et al., 2023). We have added this to the manuscript as the new Supplementary Figure 1E-F and accompanying text on page 5: ***“Fibrosis in Atg7Ad gWAT was obesity-dependent, as control (NCD)-fed mice did not develop fibrotic adipose tissue (Fig S1E-F).”***

NEW Supplementary Figure 1E-F: E) Quantification of picosirius red positive area as a percentage of the total area of stained gWAT. $n = 8-9$ mice. F) Picosirius red staining (PSR), specifically staining collagen I and III, of gWAT depots harvested from NCD-fed WT and Atg7Ad mice after 16 weeks of feeding. Scale bar, 200 μm .

Importantly, are xanthine or hypoxanthine levels elevated at this 9-week timepoint?

To directly address this, we now measured serum levels of hypoxanthine and xanthine at the 9-week timepoint in Atg7Ad and WT mice. We found that hypoxanthine and xanthine were not accumulated in the serum of Atg7Ad mice compared to WT mice at this time point (Reviewer Figure 2A). While systemic accumulation in serum was not observed, we hypothesise that localised changes in xanthine and hypoxanthine concentration within the adipose tissue microenvironment contribute to fibrosis development and are not reflected in the serum yet. Such localised metabolite conditions could influence nutrient competition and macrophage polarisation in a spatially-restricted manner. In line with this, our existing data (Fig. 6) show that controlled in vitro treatment of macrophages with xanthine and hypoxanthine induces a tissue-repair/pro-fibrotic phenotype. In addition, newly obtained data also demonstrate that macrophage polarization in gWAT is pushed towards MHCII^{low} tissue repair phenotype after 9 weeks of HFD (Reviewer Figure 2B). Although the precise kinetics of this process in vivo remain difficult to resolve, our findings suggest that even moderate, localised increases in xanthine or hypoxanthine could be sufficient to initiate fibrotic responses.

Reviewer Figure 2: A) Serum hypoxanthine and xanthine levels are comparable between WT and Atg7Ad obese mice at an early obesity timepoint. Concentration of serum xanthine and hypoxanthine in WT and Atg7^{Ad} mice fed with HFD for 9 weeks. $n = 7-11$ mice. B) Quantification of

MHCII^{low} macrophages by flow cytometry presented as the frequency of total macrophages. n = 5 mice. Statistical analysis by unpaired t-test.

2. In Figure 1C, it's shown that autophagy flux is decreased at 60 weeks of HFD. Does this timepoint correlate with the amount of fibrosis in the adipose tissue; i.e. does fibrosis increase when autophagy decreases?

We thank the reviewer for raising this interesting question. To address this, we assessed the levels of fibrosis by picrosirius red staining in gWAT from mice fed NCD or HFD for 10, 16, 30, and 60 weeks. Our data show a notable increase in fibrosis from 16 weeks of HFD feeding (new Supplementary Figure 1G). We found that fibrosis significantly increases and then persists during HFD feeding. This is consistent with clinical observations in humans, where fibrosis is generally considered irreversible once established. In contrast, autophagy flux shows a dynamic pattern, with initial upregulation during early HFD exposure (16 and 30 weeks) and a subsequent decline by 60 weeks (Fig. 1C). This reflects autophagy's dynamic and adaptive nature that fluctuates in response to cellular needs. Since fibrosis is a largely unidirectional and cumulative process, a direct temporal correlation of an increase in fibrosis with the decrease in autophagy flux at 60 weeks is difficult to establish. However, since the observed reduction in autophagy flux at 60 weeks coincides with an increase in fibrosis, the decline in autophagy may contribute to the maintenance of fibrosis during the later stages of HFD-induced obesity. It is also worth noting that the level of fibrosis observed in HFD-fed WT gWAT remains significantly lower than that seen in Atg7Ad mice under HFD (Fig. 1F), further supporting the protective role of autophagy against excessive fibrotic deposition. This figure has now been added to the manuscript as Supplementary Figure 1G, together with the following text on page 5: ***"In WT mice, we noted that fibrosis significantly increased between 16 and 30 weeks of HFD feeding and then persisted over chronic HFD exposure up to 60 weeks (Fig S1G), suggesting that a decline in autophagy at 60 weeks may contribute to the maintenance of fibrosis."***

G) picrosirius red staining (fibrosis)

NEW Supplementary Figure 1G: PSR staining of gWAT depots harvested from NCD and HFD-fed WT mice after 10, 30, and 60 weeks of feeding. Scale bar, 200 μ m.

3. The authors present interesting data that show despite increased gWAT fibrosis, glucose homeostasis is significantly improved in *Atg7*KO mice, perhaps due to improved fat storage in the subcutaneous depot. It's been shown that macrophages in this depot also significantly affect adipose tissue function. How does *Atg7*KO affect the subcutaneous depot with regard to xanthine/hypoxanthine secretion and macrophage phenotypes?

The reviewer raises a very valid point here. Like the reviewer, we believe that glucose homeostasis is impacted by the increased subcutaneous fat storage (iWAT) in *Atg7^{Ad}* mice, as these have commonly been linked (McLaughlin et al., 2011, Jocken et al., 2018, Booth et al., 2018). We have now examined iWAT more closely and found that – in contrast to gWAT – there is no increase in $F4/80^+ CD64^+$ macrophage accumulation (new Fig S6A). In addition, we found that this fat depot does not display an increased proportion of tissue-reparative macrophages defined by $MHCII^lo Lyve1^{hi}$ markers (new Fig S6B). These data suggest that macrophage phenotypes notably differ between iWAT and gWAT depots. In addition, similar observations between the depots were previously made in relation to pro- vs anti-inflammatory markers (Kralova Lesna et al., 2016, Giron-Ulloa et al., 2020, Strand et al., 2022). Furthermore, we found that xanthine and hypoxanthine secretion from iWAT adipocytes was significantly increased in *Atg7^{Ad}* mice (new Fig S6C), however, the secreted amount was approximately 14-fold lower compared to secretion from gWAT adipocytes (Fig 3D).

These data are now shown in Supplementary Figure 6 and the following text was added to the manuscript on page 10: ***“In contrast to gWAT, iWAT displayed no increase in F4/80+ CD64+ ATM or***

Lyve1^{high} MHCII^{low} PVM accumulation, suggesting that macrophage abundance and phenotypes notably differ between the depots in Atg7Ad mice (Fig S6A-B). A significant, but much lower secretion of xanthine and hypoxanthine was secreted from Atg7Ad iWAT adipocytes compared to gWAT (Fig 3D and Fig S6C). This further underscored the intrinsic differences between these depots.”

NEW Supplementary Figure 6: Atg7Ad iWAT lacks macrophage accumulation and shows modest but significantly increased xanthine and hypoxanthine secretion. A) Flow cytometry analysis of F4/80⁺ CD64⁺ macrophage number in iWAT. n = 5-6 mice. B) Quantification of perivascular macrophages identified as MHCII^{low} Lyve1^{high} by flow cytometry presented as frequency of total macrophages. n = 7-11 mice. C) Concentration of xanthine and hypoxanthine secreted from iWAT adipocytes cultured over 24 hours ex vivo. Adipocytes were isolated from WT and Atg7^{Ad} mice fed with HFD for 16 weeks. n = 8-10 mice. Statistical analysis by unpaired t-test (A-C).

4. Lastly, in Figure 3G, it's shown that overnight treatment of adipose explants with the apoptosis inhibitor does not affect secretion of xanthine/hypoxanthine. Overnight treatment is not a long time frame, especially in the context of 16 weeks of HFD feeding, thus the effect of this treatment could be underestimated. As a counter experiment, it would be useful to see if inducing apoptosis can increase xanthine/hypoxanthine secretion in explants to rule out apoptosis as a contributing factor to the xanthine secretion observed in Atg7KO adipocytes.

We thank the reviewer for the excellent suggestion, which we have now experimentally addressed. To evaluate whether inducing apoptosis can increase xanthine/hypoxanthine secretion, we treated gWAT explants from HFD-fed mice with the apoptosis inducer staurosporine (10 μM) for 24 hours in vitro. This treatment resulted in an increase in cleaved caspase-3 compared to total caspase-3 (new Fig. S3G-H), a marker of apoptosis (Porter and Jänicke, 1999). The corresponding western blot and quantification are shown. However, we found that triggering apoptosis did not alter the levels of xanthine and hypoxanthine secretion in either obese WT or Atg7Ad gWAT explants (new Fig. 3H). These outcomes are consistent with Q-VD-OPh apoptosis inhibition already reported in the manuscript, and suggest that apoptosis does not contribute to the secretion of xanthine and hypoxanthine from adipocytes in Atg7Ad gWAT.

The figures below have now been added to the manuscript, together with the following text on page 7: ***“To exclude that increased apoptosis of Atg7Ad adipocytes is the reason for elevated nucleoside production and/or secretion, we inhibited gWAT apoptosis by pan-caspase inhibitor Q-***

VD-Oph or induced gWAT apoptosis by staurosporine (STS) (Fig S2G-H and S3G-H). Neither treatment had an impact on xanthine and hypoxanthine release from WT or Atg7Ad adipocytes (Fig 3G-H).... These results suggest that the mild increase in apoptosis due to loss of autophagy does not contribute to nucleoside secretion. ... Taken together, these assays suggest that while autophagy is not essential for efflux, it limits the excessive release of xanthine and hypoxanthine from visceral adipocytes during obesity.” In addition, the following was added to the Materials and Methods section on page 24: “To determine the contribution of apoptosis, WAT explants were incubated in full RPMI supplemented with either DMSO (vehicle control) or 20 μ M Q-VD-Oph overnight or 10 μ M staurosporine for 24 hours.”

H)

NEW Figure 3H: Concentration of secreted xanthine and hypoxanthine after treatment of gWAT explants with either DMSO or 10 μ M staurosporine (STS), an apoptosis inducer, for 24 hours ex vivo. gWAT was isolated from WT and Atg7Ad mice fed with HFD for 16 weeks. n = 11 mice.

G)

H)

NEW Supplementary Figure 3G-H: WT and Atg7^{Ad} mice were fed HFD for 16 weeks. gWAT explants were cultured over 24 hours ex vivo while treated with either DMSO or 10 μ M staurosporine (STS), an apoptosis inducer. G) Representative western blot analysis of total caspase 3, cleaved caspase 3, and vinculin in gWAT. H) Cleaved caspase 3 to measure apoptosis was quantified from blots in (G). n = 8-9 mice. Statistical analysis by two-way ANOVA with Šídák multiple comparisons test (H).

5. Minor comment: could the authors comment on how a blockade in autophagy could lead to increased secretion of these metabolites?

We show that blockage in autophagy leads to increased intracellular accumulation of xanthine and hypoxanthine (Fig 2D, J). We believe that the increased secretion of these metabolites reflects intracellular concentrations (Fig 3C-D), but that the release itself is independent of autophagy. Autophagy-dependent intracellular rewiring has been previously reported to impact both tissue and body-wide outcomes (Poillet-Perez et al., 2018; Richter et al., 2023). Notably, release of nucleosides and nucleoside derivatives, including hypoxanthine, inosine, and uridine has been identified as a consequence of cellular stress in multiple studies (Fan et al., 2019, Deng et al., 2018, Pfeifer et al., 2024). In line with this, we found that autophagy depletion in adipocytes resulted in the upregulation of stress response pathways (Fig S2A). Moreover, Cai et al. (2018) have previously shown that HFD leads to cellular stress which is regulated in an autophagy-dependent manner. Therefore, we believe that a blockade in autophagy leads to a) increased accumulation of xanthine and hypoxanthine due to metabolic rewiring and b) release of these metabolites as a consequence of intracellular stress induction, which must be conveyed to the microenvironment to maintain tissue homeostasis by means of extracellular signalling. This hypothesis has been further elaborated, and we provide more evidence on this point in response to Reviewer 3. We have now added this to the discussion on page 14 as: ***“Our data show that the release of xanthine and hypoxanthine occurs independently of autophagy, but instead autophagy-dependent metabolic rewiring results in their increased intracellular accumulation. We uncover for the first time that autophagy acts as a brake on the active accumulation of nucleosides and nucleobases which are subsequently released, either actively or passively.”***

Reviewer #3 (Remarks to the Author):

In this study Piletic et al. report that loss of Atg7 (an autophagy effector) in white adipose tissue leads to 1) fibrosis, 2) macrophage infiltration, 3) release of purine catabolites from adipocytes in the setting of high-fat diet. The authors utilize conditional mouse model for Atg7ko, tissue explants, cultured primary adipocytes and conditioned medium from the adipocytes. Links between autophagy and fibrosis and macrophages are known.

We respectfully disagree with the reviewer on this point, as the role of adipocyte autophagy in shaping macrophage phenotype in adipose tissue during high-fat diet remains largely unexplored, with existing studies lacking mechanistic insight and showing conflicting results, particularly in the context of adipose tissue fibrosis.

The novelty of the manuscript is the link between adipose autophagy and purine metabolism, which is also the facet of the manuscript that the authors could have investigated further.

Specific comments

1) Why does hypoxanthine/xanthine accumulate intracellularly, and why are they secreted more when autophagy is blocked in adipocytes? What is the underlying mechanism. This is a key missing point in the manuscript.

We thank the reviewer for their comment. As pointed out in a response to reviewer no. 2, we hypothesize that the loss of autophagy triggers the production of xanthine and hypoxanthine and

their accumulation intracellularly, which then get released in an autophagy-independent way. Here is additional evidence to support these statements.

A) We provide evidence that loss of autophagy leads to intracellular accumulation of hypoxanthine and xanthine with the following experiments already shown in the original manuscript: 1) The adipocyte proteomics and metabolomics data demonstrate the upregulation of regulators of the pentose phosphate pathway and downstream purine metabolism, the end products of which are xanthine and hypoxanthine. 2) We demonstrate that these products are formed through the conversion of inosine via purine nucleoside phosphorylase (PNP) (Fig. 3F). While the pentose phosphate pathway supports de novo purine synthesis by generating ribose-5-phosphate and NADPH (Ferrier, 2014), stress conditions shift its balance toward purine catabolism (Fan et al., 2019). Under such conditions, increased nucleotide turnover and reduced recycling efficiency can lead to the accumulation of inosine, which is then metabolised by PNP into hypoxanthine and xanthine. This accumulation has been reported under various stress conditions (Kather et al., 1990; Guo et al., 2016; Fromme et al., 2018; Fan et al., 2019; Pfeifer et al., 2024). Since loss of autophagy commonly leads to a reduced ability of the cell to mitigate stress (Kroemer et al., 2010), it is likely that the stress and metabolic rewiring we observe upon loss of autophagy in Atg7Ad adipocytes further exacerbate the accumulation of xanthine and hypoxanthine.

B) To test whether xanthine and hypoxanthine levels found in the extracellular milieu are originating from these increased intracellular levels in adipose tissue, we lysed gWAT explants with 0.1 % Triton-X for 1 h before the end of incubation (i.e. after 23 h). Full lysis led to a significant change in xanthine and hypoxanthine release, which was increased in both WT and Atg7^{Ad} gWAT explant supernatants (new Fig 3I). Importantly, the full lysis-induced secretion of xanthine and hypoxanthine was much higher from Atg7^{Ad} gWAT. These results, together with our previous observations, suggest that the loss of autophagy in adipocytes triggers intracellular accumulation of xanthine and hypoxanthine, which are released from the cell in an autophagy-independent way. Whether the release is through passive or active secretion is difficult to discern with the ex vivo material at our disposal and goes beyond the scope of this study in which we set out to illuminate the role of autophagy. The figure below and the text were added to the manuscript on page 7 as: ***“However, when gWAT explants were lysed with 0.1% Triton-X, this resulted in markedly elevated extracellular levels in Atg7Ad explants (Fig 3I). ... Furthermore, autophagy-depleted adipocytes actively generate xanthine and hypoxanthine through PNP activity that can be released when cells are fully lysed. Taken together, these assays suggest that while autophagy is not essential for efflux, it limits the excessive release of xanthine and hypoxanthine from visceral adipocytes during obesity.”***

Some sections were revised to clarify that release occurs independently of autophagy, but autophagy acts to limit the extent of their intracellular accumulation

- Page 2, abstract: ***“We uncover that autophagy restrains purine nucleoside metabolism in obese adipocytes. This ultimately leads to a reduced release of the purine catabolites xanthine and hypoxanthine.”***
- Page 11, paragraph 2: ***“While the release is autophagy-independent, autophagy controls the purine nucleoside metabolism in adipocytes. Dysregulation of this metabolic pathway***

leads to excessive nucleoside release, which in turn shifts the balance from tissue inflammation toward fibrosis.”

- Page 14, paragraph 1: *“Our data show that the release of xanthine and hypoxanthine occurs independently of autophagy, but instead autophagy-dependent metabolic rewiring results in their increased intracellular accumulation. We uncover for the first time that autophagy acts as a brake on the active accumulation of nucleosides and nucleobases, which are subsequently released, either actively or passively.”*

NEW Figure 3I: Concentration of released xanthine and hypoxanthine from gWAT explants after cell lysis with 0.1 % Triton-X for 1 h before the end of incubation. gWAT was isolated from WT and Atg7Ad mice fed with HFD for 16 weeks. n = 4 mice. Statistical analysis by two-way ANOVA with Šidák multi comparisons test (I).

C) Additional experiments were performed to better understand the signalling mechanism underlying hypoxanthine and xanthine accumulation upon autophagy deletion. Our working hypothesis was that loss of autophagy leads to a re-wiring of the mitochondrial metabolism and cellular stress, accumulating purine metabolites, including hypoxanthine and xanthine. Notably, conditions in which mitochondrial metabolism is impacted (e.g by inhibiting glycolysis, turning off the energy sensing activity of AMPK, or replenishing TCA metabolites) have been shown in previous studies to restore the activity of purine metabolism to normal levels (Guo et al., 2016, Fan et al., 2019). If these conditions restore purine levels in our scenario, it would suggest that the absence of autophagy leads to rewired cellular metabolism. Therefore, we selected the following pharmacological agents: 2-deoxy-d-glucose (2-DG, glycolysis inhibitor), with 10 μM AMPK inhibitor SBI-0206965 (stress signalling inhibitor) or 10 mM glutamine (supports mitochondrial metabolism and replenishes TCA cycle). Treatment of Atg7Ad obese adipocytes with 2-DG, Gln, and AMPK inhibitor, in contrast to WT obese adipocytes, led to a reduction in xanthine and hypoxanthine production, albeit not back to WT levels (Reviewer Figure 3). These data collectively demonstrate that excessive purine production is due to metabolic imbalance of the cell when autophagy is absent. Importantly, targeted metabolic interventions can attenuate the effect of metabolic rewiring, reinforcing autophagy's essential role in maintaining the purine metabolic balance. Autophagy maintains purine metabolic homeostasis by supporting mitochondrial function and restraining compensatory glycolytic flux.

Reviewer Figure 3: Xanthine and hypoxanthine accumulation in response to metabolic interventions. Concentration of xanthine and hypoxanthine accumulated from gWAT adipocytes cultured for 24 hours in full RPMI medium (CTRL), glucose-free RPMI medium supplemented with 10 mM 2-deoxy-d-glucose (2-DG), full RPMI with excess 10 mM glutamine (Gln) or full RPMI supplemented with 10 μ M AMPK inhibitor SBI-0206965 (AMPKi). gWAT was isolated from WT and *Atg7^{Ad}* mice fed with HFD for 16 weeks. n = 4 mice.

2) Perilipin1 staining is not a marker of adipocyte viability. The ‘perilipin1 negative adipocytes’ in Fig. S2C,D are presumably non-adipocytes (macrophages and other the immune cells) in the tissue. It is not clear how the authors conclude that perilipin1-negative adipocytes are in fact adipocytes. This is contradictory to what is shown in Fig. 4E wherein adipocyte numbers between the two groups remain similar.

Thank you for your comment. The expression of adipocyte-specific lipid droplet protein perilipin 1, is the most commonly reported marker of adipocyte viability by immunofluorescence in the literature. Multiple studies have addressed this experimentally, demonstrating that loss of adipocyte viability leads to degeneration of lipid droplets, which can be detected by the loss of lipid droplet-associated proteins. In contrast to non-viable adipocytes, viable adipocytes retain perilipin-1 immunoreactivity (Cinti et al., 2005, Feng et al, 2011, Wang et al., 2020). Nonviable adipocytes lose the expression of Perilipin-1 and become surrounded by macrophages, which form crown-like structures for adipocyte clearance. We agree that a limitation of this technique is that the increase in perilipin-negative cells could also be from cells other than adipocytes. Nevertheless, adipocytes can still be distinguished from other cell types in the adipose tissue due to lipid droplet formation, offering differentiation based on morphological distinction. Thus, perilipin-1-negative regions surrounded by prominent DAPI staining are morphologically consistent with adipocytes (i.e., large unilocular lipid droplets), supporting their identification as dying adipocytes rather than immune cells. We have clarified this point in the revised figure legend and manuscript text and decided to present in Fig. S2F adipocyte-specific lipid droplet protein Perilipin-1 positive adipocytes, as the protein expression is adipocyte-specific: ***“To exclude that these findings could be solely explained by increased cell death of the autophagy-deficient adipocytes, we assessed adipocyte viability by Perilipin-1 staining, a widely used method in the field (Cinti et al., 2005). Clearly and as expected, adipocyte viability was markedly reduced due to HFD feeding, while autophagy depletion only increased the cell death minimally (Fig S2E-F). Perilipin-1 staining is based on morphological***

distinction of unilocular adipocytes and thus gives limited information about the type of cell death induced by loss of autophagy.”

NEW Supplementary Figure 2E-F: Representative immunofluorescence staining of Perilipin-1 on gWAT sections from WT and Atg7Ad mice following HFD feeding for 16 weeks (E). Quantification of Perilipin-1 staining as a percentage of Perilipin-1 positive adipocytes compared to total area (F) from (E). Legend: blue = dapi, red = Perilipin-1. n = 7-8 mice, each section quantified across multiple areas of interest.

3) Many of the experiments rely on isolated adipocytes from wt and Atg7ko adipose tissue which have high cell death (Fig. S2E). Are the phenotypes of high hypoxanthine/xanthine release, macrophage infiltration, lower adipocyte markers all a result of cell death?

The reviewer is correct to point out that Atg7^{Ad} gWAT explants display a tendency towards slightly increased levels of apoptosis in Fig S2E of the manuscript (now Fig S2H). Importantly, we demonstrate that the main driver of adipocyte cell death is the diet rather than the genotype (New Supplementary Fig. 2E-F). There is a lot of published research available demonstrating that increased macrophage infiltration, macrophage polarization to a pro-inflammatory phenotype and low adipocyte markers, including perilipin-1, adiponectin, and leptin are result of adipocyte cell death which occurs during obesity (Cinti et al, 2005, Hill et al., 2018, Jaitin et al., 2019, Wang et al., 2020, Trim and Lynch, 2022). As pointed out in the experiment performed for reviewer no. 2, question no. 4 above (new Fig. 3H), newly collected data, together with Q-VD-OPh apoptosis inhibition already reported in the manuscript (Fig. 3G), suggest that apoptosis does not contribute to the secretion of xanthine and hypoxanthine. On the other hand, our results suggest that extracellular accumulation of xanthine and hypoxanthine could, in part, be due to passive release as demonstrated through detergent-induced cell lysis. We could not ascertain how the release happens in vivo, passive or active secretion might be at play or cell death that is non-apoptotic.

4) Do intracellular levels of xanthine/hypoxanthine change in Atg7ko obese adipocytes?

Yes, we see that xanthine and hypoxanthine increase in Atg7Ad adipocytes, as already included in the original manuscript under Fig 2J. Experiments provided in the original manuscript and added in

this revision demonstrate that this accumulation results from metabolic rewiring caused by autophagy loss, including increased glycolytic flux, impaired mitochondrial metabolism and enhanced stress signalling. Pharmacological interventions targeting these pathways reduced xanthine and hypoxanthine levels, highlighting autophagy's critical role in maintaining purine metabolic homeostasis by supporting mitochondrial function and preventing excessive glycolytic compensation.

5) Are the pathways shown in Fig. 1B upregulated or downregulated in obese state?

Fig. 1B shows the most significantly dysregulated pathways between lean and obese humans. Rather than classifying these pathways strictly as “upregulated” or “downregulated”, we wanted to identify which pathways were overall the most perturbed in the transition from a lean to an obese state. This approach accounts for the complexity of pathway regulation, where individual genes may have opposing roles, with some activating and others inhibiting the pathway's activity. We have clarified this in the manuscript on page 4, paragraph 2: ***“Comparison of white adipocytes between lean and obese states revealed macroautophagy as one of the most notable significantly dysregulated pathways, together with multiple pathways known to be impacted by weight gain, including insulin signalling, lipid metabolism, and tissue repair (Fig 1A-B).”***

6) Reporting: The manuscript could use a considerable effort in reporting the n values and putting more details in legends and data shown. The reporting summary says that the exact n is reported; however, the authors are urged to revisit each figure panel and be consistent in the reporting of sample size. To cite a few instances - What is the n for Fig. S2E? The data shows 4 points. The legend mentions 5-6 mice. Is this a typo? Or are all data not being reported? Similarly, what is the n for Fig. S2D? The legend mentions 7-8 mice, and the graph shows data points that are >50. What are these data points? Fields? The authors also show data from blots without showing the blots themselves.

We thank the reviewer for their careful assessment of the manuscript. We have thoroughly revisited the reporting of sample sizes and the figure legends. Regarding Fig. S2E (now Fig S2H), we would like to clarify that the number of data points corresponds correctly to the stated number of biological replicates however, due to close clustering of data points, individual points may appear visually overlapping. For Fig. S2D (now Fig S2F), we clarified that the data points represent quantified fields, not individual animals, and this is now explicitly stated in the legend. We have also clarified that proteomics data presented in Fig. S2A-D and S3D correspond to n = 4 mice. In addition, we have added the corresponding source blots where quantitative data were shown without the original images, ensuring complete transparency. For Fig 1C, we now provide a representative WB for measurement of autophagy flux under Fig S1A. For Fig S2E (now Fig S2H), where we provide quantified data for caspase 3 cleavage, we have now included the representative WB as well (Fig S2G).

NEW Supplementary Figure 1A: A) A representative western blot analysis of LC3-I, LC3-II, and vinculin in gWAT to measure autophagy flux. WT mice were fed a normal chow diet (NCD) or high fat diet (HFD) for 10, 30 and 60 weeks. An example western blot is shown for 30 weeks of altered diet.

NEW Supplementary Figure 2G: G) Cleaved caspase 3 was assessed by western blot analysis in gWAT explants cultured overnight ex vivo and treated with either DMSO or 20 μ M Q-VD-OPh, a pan-caspase inhibitor. gWAT was isolated from WT and Atg7Ad mice fed with HFD for 16 weeks. Representative western blot analysis of total caspase 3, cleaved caspase 3, and vinculin in gWAT to measure apoptosis.

Other comments

1) There is robust decrease in Atg7 in the Atg7cko (Fig. 1A). Despite this, there is considerable signal of LC3II in the Atg7cko samples (Fig. S1B). Perhaps this is due to WT and Atg7ko samples being run on separate gels. The flux calculations are valid. Separate blots make it seem that LC3II levels are similar between WT and Atg7ko mice on HFD which is most likely not the case.

In this study, we have taken advantage of a previously published and validated mouse model (Cai et al., 2018, Richter et al., 2023). Importantly, robustly decreased Atg7 levels in adipocytes are accompanied by the accumulation of LC3B-I, with a less obvious decrease in LC3B-II. This is usually found in mouse models where key autophagy genes are knocked out (for example Richter et al., 2023 - Fig 2C, Kabat et al., 2016 - Fig 1B), likely depending on the basic autophagy flux in the respective tissues. Key autophagy proteins such as Atg5, Atg7 and Atg16L1 promote the formation and extension of autophagosome membranes by conjugating LC3B to phosphatidylethanolamine, therefore, either decreased LC3B-II or increased LC3B-I levels upon BafA1 can indicate reduced flux of the autophagosomes (best is both, of course). Furthermore, contamination with non-adipocytes of the tissue may mask a good reduction of the flux. This has now been added as a limitation to the description of the figure S1C: **“Limited increase in LC3B-II is likely influenced by tissue heterogeneity and technical limitations of adipose tissue western blotting.”**

2) Is the KO specific to adipocytes? What is the status of Atg7 in other tissues?

To assess the specificity of deletion, we took advantage of *Adipoq-Cre^{ERT2}* mouse model crossed to a tdTomato reporter mouse to visualize the site of recombination when induced with tamoxifen. We found that tamoxifen-induced tdTomato expression in this model, specifically impacted adipose

tissues, including gWAT and to a lower extent BAT (Reviewer Figure 4A). We did not observe any significant recombination in non-adipose tissues, including liver, lungs and spleen, confirming that *Atg7* knockout is specific to adipocytes and adipose tissue. Analysis of *Atg7* expression across multiple tissues showed a marked reduction in *Atg7* mRNA levels in gWAT and BAT, while expression remained unchanged in the lungs and spleen (Reviewer Figure 4B). We added a comment in the manuscript on page 4 as: “*Specificity of Cre expression in adipose tissues and Atg7 expression in metabolic tissues were assessed prior to this study (Richter et al., 2023).*”

Reviewer Figure 4: Adipoq-Cre^{ERT2} drives adipose tissue-specific deletion of *Atg7*. A) Stereo-fluorescence microscopy of dTomato expression after tamoxifen administration in brown (BAT) and gonadal white (gWAT) adipose tissue, lungs and spleen. B) Relative mRNA expression of *Atg7* in gWAT, BAT, lungs and spleen of WT and *Atg7^{Ad}* mice measured by qRT-PCR. n = 3-4 mice.

REVIEWERS' COMMENTS

Reviewer #1 (Remarks to the Author):

in this revised manuscript reviewed earlier by me, authors carefully considered all of the reviewers concern, and addressed them with additional experimental data, edited the manuscript text and figures thoroughly, included limitations of the study and relevance to human, added appropriate supplementary table details (please correct the supplementary resources table with proteomics data id). It is an important work, and wish that authors peruse this work more on human.

We thank the reviewer for their thoughtful comments to improve the manuscript and for recognizing the importance of our work.

Reviewer #2 (Remarks to the Author):

The revised manuscript is much improved and addresses the points raised in my review.

We thank the reviewer for their thoughtful comments to improve the manuscript and for recognizing the importance of our work.

Reviewer #3 (Remarks to the Author):

Remaining comments:

Regarding the use of triton-X-100 to distinguish total intracellular metabolite pool, the term “secretion” should be revised to “release” or “lysis-induced release” in all places - to avoid implying an active export mechanism where none is demonstrated. The authors say as much but only in some places. Please correct.

We have revised the term “secretion” to “release” as advised to sections below:

Page 4, paragraph 1: “Failure to meet their metabolic demands in the absence of autophagy leads to elevated purine nucleoside production and release.”

Page 7, paragraph 1: “Xanthine and hypoxanthine levels more than doubled in the secretome of obese Atg7Ad adipocytes (Fig 3D), and we observed a strong negative correlation between the activity of the autophagy pathway and their release from obese gWAT in WT and Atg7Ad mice (Fig 3E). Purine nucleoside phosphorylase (PNP) plays a key role in purine catabolism, limiting the production of purine nucleobases³⁵. Inhibition of PNP activity by the clinically approved drug forodesine resulted in a significantly lower xanthine and hypoxanthine release from both WT and Atg7Ad adipocytes, with the latter being reduced to almost WT levels upon PNP inhibition (Fig 3F). To exclude that increased apoptosis of Atg7Ad adipocytes is the reason for elevated nucleoside production and/or release, we inhibited gWAT apoptosis by pan-caspase inhibitor Q-VD-OPh or induced gWAT apoptosis by staurosporine (STS) (Supplementary Fig 2G-H and Supplementary Fig 3G-H). Neither treatment had an impact on xanthine and hypoxanthine efflux from WT or Atg7Ad adipocytes (Fig 3G-H). However, when

gWAT explants were lysed with 0.1% Triton-X, this resulted in markedly elevated extracellular levels in Atg7Ad explants (Fig 3I). These results suggest that the mild increase in apoptosis due to loss of autophagy does not contribute to increased extracellular nucleoside levels.”

Page 10, paragraph 1: “A significant, but much lower release of xanthine and hypoxanthine was observed from Atg7Ad iWAT adipocytes compared to gWAT (Fig 3D and Supplementary Fig 6C).”

Fig 3:

“C) Relative abundance of nucleosides released by gWAT adipocytes isolated from WT and Atg7Ad mice following HFD feeding for 16 weeks and measured in metabolomics analysis. n = 3 mice.

D) Concentration of xanthine and hypoxanthine released from gWAT adipocytes cultured over 24 hours ex vivo. Adipocytes were isolated as in (C). n = 5 (WT) or 6 (Atg7Ad) mice.

E) Correlation analysis of the level of autophagy flux in gWAT and concentration of xanthine and hypoxanthine released from gWAT adipocytes as in (D). Western blot analysis of autophagy flux was calculated as (LC3-II (Inh) – LC3-II (Veh)). n = 9 mice.

G) Concentration of xanthine and hypoxanthine released from gWAT explants cultured overnight ex vivo and treated with either DMSO or 20 μ M Q-VD-OPh, a pan-caspase inhibitor. n = 5 (Atg7Ad) or 6 (WT) mice.”

Regarding Fig. S1C, reporting samples on separate blots sets a wrong precedent for future readers. The limitation is not methodology of western blotting in adipose tissue, but your inertia of rest in not running all samples in one blot as previously requested. Stating “Limited increase in LC3B-II is likely influenced by tissue heterogeneity and technical limitations of adipose tissue western blotting.” is erroneous because the limitation is the fact that they are on separate blots and not what is stated. All the other blots reported evidently do not have those limitations.

We appreciate the reviewer’s comment regarding Supplementary Figure 1C. In response, we have replaced the previously reported separate blots with a new blot that includes all samples run on a single gel, in line with the reviewer’s suggestion. Accordingly, we have also removed the statement “Limited increase in LC3B-II is likely influenced by tissue heterogeneity and technical limitations of adipose tissue western blotting” from the Supplementary Figure 1C description, as it is no longer applicable.